# Dedicated surveillance mechanism controls G-quadruplex forming non-coding RNAs in human mitochondria

Zbigniew Pietras[1,2], Magdalena A. Wojcik[1,3], Lukasz S. Borowski [1,3], Maciej Szewczyk [1,3], Tomasz M. Kulinski [1], Dominik Cysewski [1], Piotr P. Stepien[1,3], Andrzej Dziembowski [1,3] & Roman J. Szczesny [1,3]

The GC skew in vertebrate mitochondrial genomes results in synthesis of RNAs that are prone to form G-quadruplexes (G4s). Such RNAs, although mostly non-coding, are transcribed at high rates and are degraded by an unknown mechanism. Here we describe a dedicated mechanism of degradation of G4-containing RNAs, which is based on cooperation between mitochondrial degradosome and quasi-RNA recognition motif (qRRM) protein GRSF1. This cooperation prevents accumulation of G4-containing transcripts in human mitochondria. In vitro reconstitution experiments show that GRSF1 promotes G4 melting that facilitates degradosome-mediated decay. Among degradosome and GRSF1 regulated transcripts we identified one that undergoes post-transcriptional modification. We show that GRSF1 proteins form a distinct qRRM group found only in vertebrates. The appearance of GRSF1 coincided with changes in the mitochondrial genome, which allows the emergence of G4-containing RNAs. We propose that GRSF1 appearance is an evolutionary adaptation enabling control of G4 RNA.

---

[1] Institute of Biochemistry and Biophysics Polish Academy of Sciences, Laboratory of RNA Biology and Functional Genomics, Pawinskiego 5A, 02-106 Warsaw, Poland. [2] International Institute of Molecular and Cell Biology, Laboratory of Protein Structure, Ks. Trojdena 4, 02-109 Warsaw, Poland. [3] Faculty of Biology, Institute of Genetics and Biotechnology, University of Warsaw, Pawinskiego 5A, 02-106 Warsaw, Poland. Correspondence and requests for materials should be addressed to A.D. (email: andrzejd@ibb.waw.pl) or to R.J.S. (email: rszczesny@ibb.waw.pl)

Human mitochondrial genome (mtDNA) is a circular DNA molecule encoding 2 rRNAs, 22 tRNAs, and 13 protein-coding genes[1]. All other proteins, including those needed for mitochondrial genome expression and RNA decay, are nuclear-encoded, synthesized in the cytoplasm, and imported into mitochondria[2].

Transcription of both mtDNA strands starts in the D-loop region and tRNAPhe gene, and continues through almost the entire genome[1–3]. As a result, long polycistronic transcripts are synthesized that undergo endonucleolytic processing that releases immature tRNAs, rRNAs, mRNAs, and non-coding RNAs (ncRNA)[4–6]. Subsequently, immature functional RNAs are modified. Messenger RNAs are oligo- and polyadenylated[7], whereas rRNAs are subject to methylation and pseudouridylation[8] and tRNAs undergo several types of nucleotide modifications and addition of CCA at the 3′ end[9].

A peculiar feature of many mitochondrial genomes, including that of humans, is an unequal distribution of guanines between DNA strands. Due to this asymmetry, the strands can be separated using buoyant density ultracentrifugation to isolate the heavy (H-strand) and light strand (L-strand). In humans, most mitochondrial genes are encoded by the G-poor strand (L-strand), thus the complementary H-strand is used as a template for transcription[1]. In contrast, RNAs derived from L-strand transcription are mostly non-coding/non-functional. Around 90% and 9% of L-strand and H-strand transcripts, respectively, constitute non-coding RNAs, which are antisense to functional RNAs transcribed from the opposite strand. Overall, steady-state levels of non-coding mtRNAs are extremely low, although transcription of the L-strand, which is the major source of mt-ncRNA, is more frequent than that of the H-strand[3,10]. Interestingly, G-rich L-strand transcripts are theoretically prone to form non-canonical four-stranded structures called G-quadruplexes (G4s)[11]. G4s present in nuclear-encoded RNAs play important roles in regulation of splicing, polyadenylation, RNA turnover, translation, and mRNA targeting[12]. The role of G4s in mtRNA metabolism is less known, although there is an evidence indicating that G4 formation during transcription of the CSBII region terminates RNA synthesis[13].

The mechanism by which human mitochondrial RNA (mtRNA) is degraded was unclear for many years. To date, the mitochondrial degradosome, a complex of the RNA helicase hSuv3[14–16] (also known as SUPV3L1) and the ribonuclease PNPase[17–19] (also known as PNPT1) is the best-described mtRNA degradation machinery in human cells[20,21]. This degradosome complex is an exoribonuclease that degrades RNA from the 3′ to 5′ end[20]. In vivo, the degradosome forms in distinct areas of mitochondria named D-foci, which co-localize with mtRNA granules (MRG) and mitochondrial nucleoids[21]. The activity of the degradosome was shown to be required for mt-mRNA turnover[21,22], 16S rRNA decay[23], and exonucleolytic processing of the 3′-end of ND6 mRNA[24]. Many mt-ncRNA species can be detected only when degradosome activity is inhibited[16,21]. This rapid disappearance of non-coding antisense mtRNAs appears to be physiologically important given their potential to interfere with the functions or expression of sense transcripts. However, the mechanism by which this highly efficient degradation is achieved is not fully understood, especially that many of mt-ncRNAs could form G4s, which can hamper transcripts degradation.

Many RNA-related mechanisms, including RNA decay, are dependent on RNA-binding proteins (RBPs). Several mitochondrial RBPs involved in different RNA-related activities have been identified[25]. One mitochondrial RBP, GRSF1, was originally described as a cytoplasmic poly(A) + mRNA binding protein[26], but was later found to localize to mitochondria[27,28].

GRSF1 belongs to a protein family characterized by the presence of a quasi-RNA recognition motif (qRRM)[29] that seems to be absent in bacteria from which mitochondria originated. This is in contrast to proteins which contain classical RRM domain represented in mitochondria by SLIRP[22].

Here we show that GRSF1 positively regulates degradosome-dependent decay of non-coding mitochondrial transcripts that form G4 structures. Comprehensive molecular analyses revealed that GRSF1 melts G4 RNA structures in mtRNAs, which facilitates their degradation by the hSuv3–PNPase complex both in vitro and in vivo. Based on phylogenetic analyses, we propose that the appearance of GRSF1 in mitochondria is an evolutionary adaptation that occurred when mitochondrial genomes underwent a transition from G4-poor to G4-rich and enabled control of mitochondrial G4 RNA levels.

## Results

**GRSF1 and the degradosome co-purify from mitochondria**. To identify potential degradosome partners, we performed co-purification studies (Fig. 1a) using human 293 cells stably expressing hSuv3 or PNPase with a C-terminal TAP tag or TAP tag fused to a mitochondria targeting sequence (control bait)[16]. Mitochondria were isolated from the cells, lysed, and the protein extracts were subjected to affinity chromatography. Co-purified proteins were identified by mass spectrometry and their amounts were quantified by a label-free approach (Supplementary Data 1). We identified seven proteins that co-purified with hSuv3 and PNPase and showed >3 mean enrichment for both baits, which we regarded as putative degradosome-associated proteins (Supplementary Data 1, sheet "Final list"; see the Methods section for details concerning the analysis criteria). Notably, PNPase was the best hit when hSuv3 was used as a bait and vice versa, thus confirming successful purification of the degradosome (Fig. 1a). Among putative degradosome-associated proteins, we found two well-documented mitochondrial RBPs—LRPPRC and GRSF1—as well as C1QBP known to be required for mitochondrial translation[30]. LRPPRC was previously proposed to suppress degradosome-mediated mtRNA decay[22], but the role of GRSF1 in degradosome-mediated RNA transactions was unknown. Pull-down experiments performed in the presence of RNase A indicated that GRSF1 associates with the degradosome in RNA-dependent manner (Supplementary Fig. 1a).

We next investigated whether GRSF1 co-localizes with the degradosome. The complex was visualized using a previously described bimolecular fluorescence complementation (BiFC) approach[21] whereas immunofluorescent labeling was applied to detect GRSF1. We found that 41% of degradosome-containing foci colocalized with GRSF1 when catalytically active degradosome was analyzed (Fig. 1b, c). Interestingly, the co-localizing fraction markedly increased to 78% when inactive degradosome was examined (Fig. 1b, c). The obtained results support the conclusion that GRSF1 associates with the degradosome. Moreover, based on our previous findings that inactive degradosomes stall on RNA substrates[21], these results strongly suggested that the GRSF1 may cooperate with the hSuv3–PNPase complex when enhancement of an activity of the complex is required.

**GRSF1 or the degradosome dysfunction increases L-strand RNAs**. To examine whether GRSF1 cooperates with the degradosome in mtRNA metabolism, we analyzed the global impact of the degradosome and its putative partner GRSF1 on the mitochondrial transcriptome. We expected that functional interactions between GRSF1 and the degradosome should manifest in a common molecular RNA phenotype caused by inactivation of GRSF1 or the degradosome. We performed

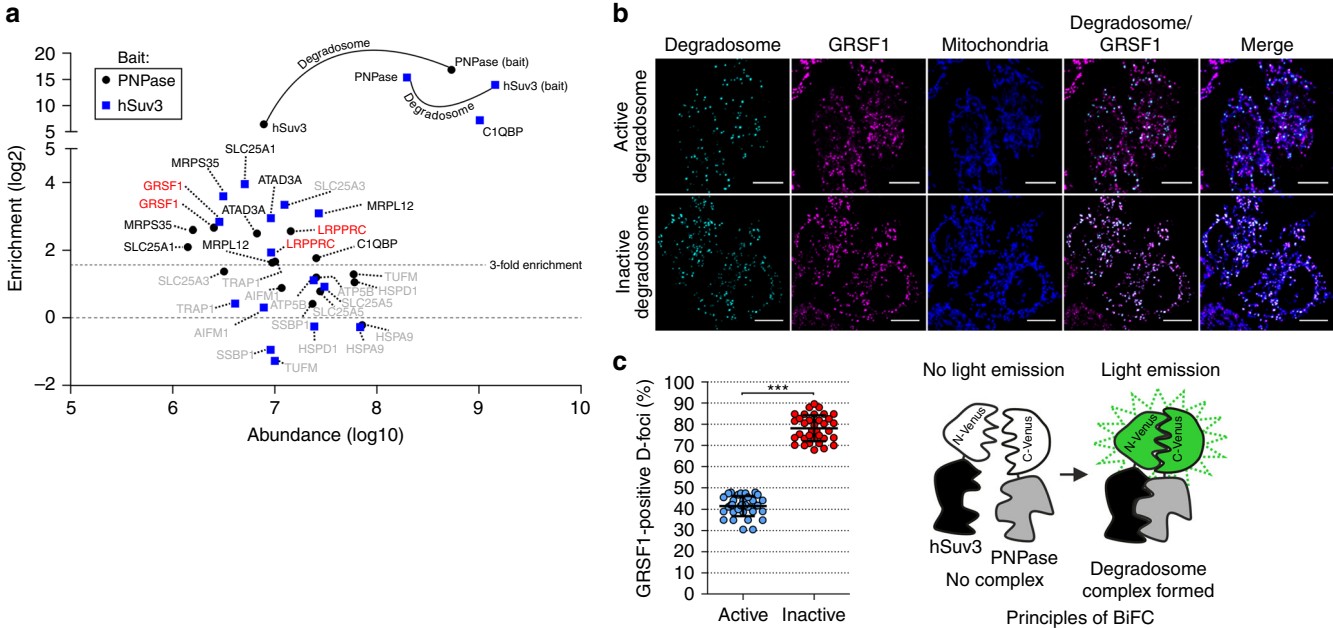

**Fig. 1** GRSF1 associates with the degradosome. **a** Mitochondrial proteins identified in all three purifications of hSuv3 or PNPase. Proteins that did not meet the criteria for consideration as putative degradosome-associated are labeled in gray. **b** Colocalization studies of the degradosome and GRSF1. The degradosome was visualized using a bimolecular fluorescence complementation (BiFC) approach, the principles of which are illustrated schematically. Catalytically active or inactive (hSuv3-G207V, PNPase-R445ER446E) components of the degradosome were transiently expressed in fusion with a split Venus protein. D-foci are not present in every cell due to transfection efficiency below 100%. GRSF1 was detected using immunofluorescence labeling. Specificity of antibodies for GRSF1 was confirmed by their pre-incubation with purified GRSF1 (Supplementary Fig. 1b). Mitochondria were stained with MitoTracker. Scale bar represents 10 μm. **c** Quantitative analysis of fluorescence microscopy data wherein 35 randomly selected cells were analyzed. Fraction of D-foci co-localizing with GRSF1 was calculated using the object-based colocalization approach. Individual values are shown. Horizontal lines represent mean values. Error bars represent standard deviation. A two-tailed unpaired t-test was applied (***P < 0.0001)

RNA-seq experiments using RNA isolated from cells in which GRSF1 or degradosome function was impaired. Dysfunction of hSuv3 was obtained using inducible expression of a dominant-negative mutant form of this protein (hSuv3-G207V), which produces an effect equivalent to that of hSuv3 silencing[16], whereas PNPase or GRSF1 levels were depleted by siRNA transient transfections (Fig. 2a–d, orange panels). As controls, we analyzed untreated cells, cells transfected with non-targeting siRNA, and cells expressing an siRNA-insensitive version of PNPase (rescue experiment) (Fig. 2a–d, green panels). Western blot analysis confirmed depletion of PNPase or GRSF1 and transgene expression (Fig. 2e).

To cover a wide range of RNA classes, we generated two types of strand-specific RNA-seq libraries. A ligation-based approach was used to analyze short RNAs (from ~20 to ~200 nucleotides), whereas longer RNAs (>~100 nucleotides) were analyzed after random fragmentation followed by reverse transcription primed with random oligomers.

Mapping of sequencing reads of long RNA libraries to the mitochondrial genome (Supplementary Fig. 2) revealed that degradosome dysfunction resulted in massive accumulation of L-strand transcripts (Fig. 2a, b), which was also apparent when the total coverage of reads mapped to mtDNA strands was calculated (Fig. 2f). Distribution of mapped reads indicated that the D-loop region was the most affected by degradosome dysfunction (Fig. 2a, b). Quantitative analysis of this region showed a 13- and 47-fold increase in the number of reads representing L-strand transcripts in PNPase-depleted and hSuv3-G207V-expressing cells, respectively, relative to untreated cells. Notably, hSuv3 dysfunction had a stronger effect than did PNPase silencing. This result is likely due to the more efficient

and homogeneous inactivation of the degradosome in hSuv3-G207V-expressing cells. All 293-hSuv3-G207V cells express the mutated form of hSuv3[16], whereas reduction of PNPase levels achieved by transient siRNA transfection can vary from cell to cell. Importantly, changes in the mitochondrial transcriptome following PNPase silencing were rescued by expression of its transgenic siRNA-insensitive counterpart (Fig. 2a, b), which confirms that the effects can be attributed to PNPase depletion.

Silencing of GRSF1 resulted in clear accumulation of L-strand transcripts (Fig. 2c, d, g); however, the extent to which the transcripts accumulated was weaker than in the case of degradosome-impaired cells. The effect of GRSF1 silencing was most pronounced in the D-loop region and was also evident for L-strand transcripts from another genomic region encompassing genes encoding CytB, ND6, ND5, and a part of ND4 (Fig. 2c). L-strand transcription within this region leads to synthesis of ND6 mRNA and long non-coding RNAs (antisense of mRNA ND5, CytB, and ND4)[21,31]. These transcripts are upregulated in degradosome-deficient cells as revealed by RNA-seq (Fig. 2a) and northern blot analyses[21]. Moreover, previous reports indicated that ND6 mRNA and the above-mentioned long non-coding RNAs are preferentially bound by GRSF1[27,32]. Because previous nucleotide-resolution analysis of GRSF1 RNA targets was not strand-specific[32], we re-analyzed eCLIP data obtained by the ENCODE initiative that were deposited in a public repository[33]. The strand-specific landscape of GRSF1 RNA targets shows that GRSF1 preferentially binds L-strand transcripts relative to H-strand counterparts (Fig. 2h). Our analysis suggested that the L-strand transcripts of the D-loop region are probably the main targets of GRSF1 as this region showed the highest enrichment among GRSF1 substrates (Fig. 2h, i). This result is in

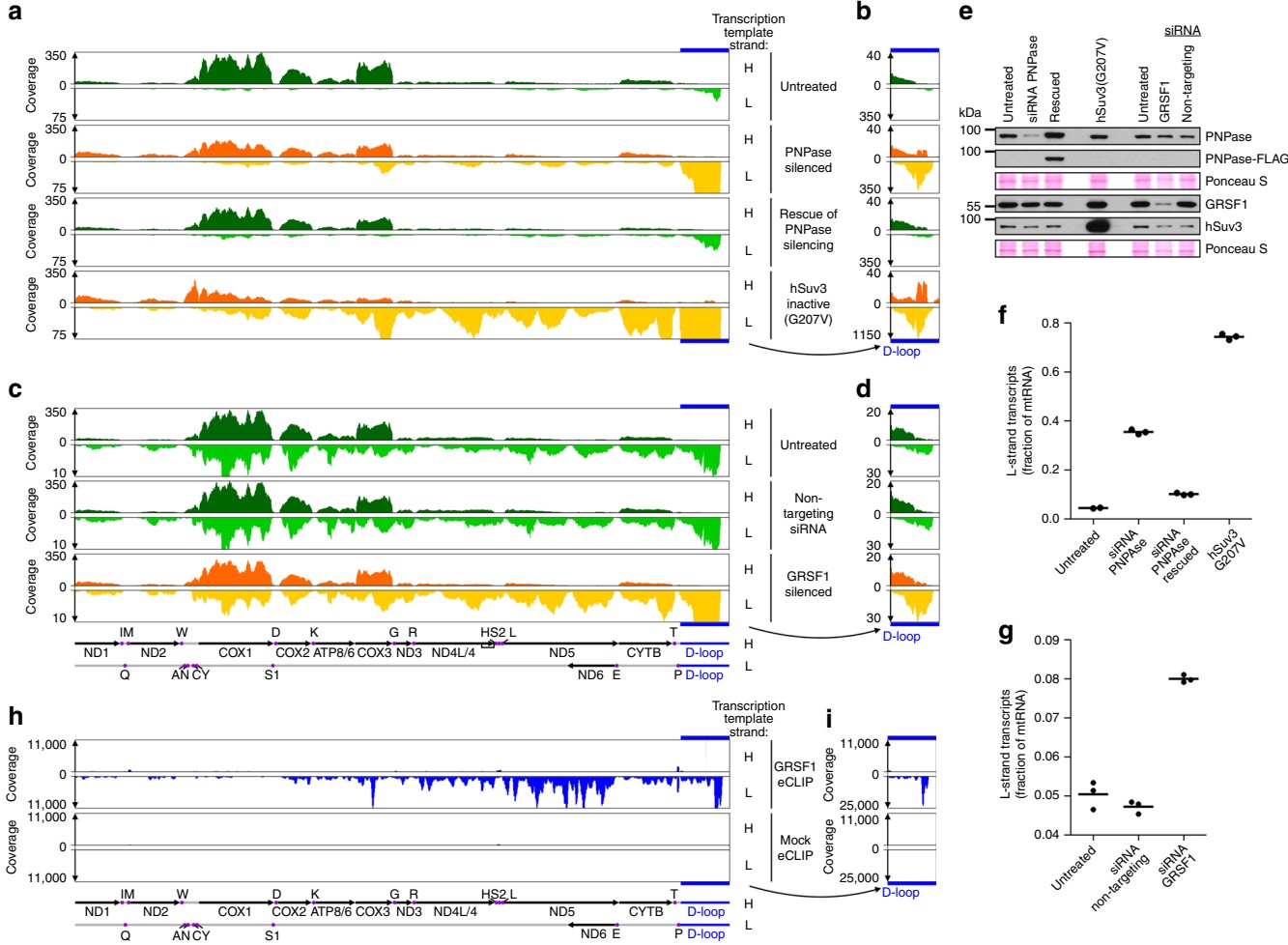

**Fig. 2** Dysfunction of the degradosome or GRSF1 upregulates RNAs originating from L-strand transcription. **a–d** Tracks of mapped reads of long transcript libraries. RNA from two set of experiments (**a–b** and **c–d**) performed in triplicate was analyzed. Panels **b** and **d** show the D-loop region with an adjusted scale. siRNA silencing of endogenous PNPase expression was rescued by expression of an exogenous siRNA-insensitive version of PNPase-FLAG. tRNA genes are labeled with single-letter symbols. S1 and S2 correspond to tRNA-Ser(UCN) and tRNA-Ser(AGY), respectively. **e** Western blot analysis of indicated proteins. Exogenous, siRNA-insensitive PNPase was detected with anti-FLAG antibodies. Membranes were stained with Ponceau S as a loading control. **f**, **g** Contribution of L-strand templated transcripts to the mitochondrial transcriptome. Individual values are shown. Horizontal lines represent mean values. **h–i** Distribution of GRSF1-binding sites across mitochondrial transcripts. eCLIP data from ENCODE database were re-analyzed. The D-loop region is shown with an adjusted scale (**i**)

agreement with our RNA-seq results (Fig. 2c, d) showing that this pool of transcripts had the highest upregulation in the presence of GRSF1 silencing.

Together, our results indicate that the degradosome is critical for shaping the transcriptomic outcome of the non-coding regions of mtDNA. Moreover, a distinct pool of mtRNAs, namely L-strand transcripts, appear to be targeted both by GRSF1 and the hSuv3–PNPase complex, thus implying their cooperation in mtRNA degradation. The finding that RNAs that are preferentially bound by GRSF1 are primary targets of the degradosome suggests that the RNA binding activity of GRSF1 could be involved in degradosome-mediated RNA decay.

**Identification of GRSF1/degradosome sensitive short RNAs.** In addition to the above-mentioned analysis of long mitochondrial transcripts we also analyzed short mtRNAs. We took advantage of the fact that libraries were sequenced in pair-end mode. For analysis, we considered only pair-end reads in the case of which sequences from both termini showed overlap of at least five nucleotides. This condition allowed us to filter data and focus on full-length RNA species. Because the applied library preparation

procedure did not include steps that are essential for reliable analysis of tRNAs[34], we did not subject this class of transcripts to quantitative analysis.

We detected a strong accumulation of a novel short RNA species in PNPase- and hSuv3-defective cells which was also upregulated to some extent in GRSF1 silenced cells (Fig. 3a, c, red dashed rectangle). This 73 nucleotide transcript originated from transcription of the non-coding strand of ND4 gene (L-strand, nucleotides 11900–11972, NC012920.1). Detailed bioinformatic analysis revealed that this transcript is post-transcriptionally modified by the addition of a non-templated CCA sequence at the 3′ end, like in the case of tRNAs, and has the potential to form secondary structures (Supplementary Fig. 3). Therefore, we named this transcript tRNA-like. RNA-seq showed that the tRNA-like transcript was detectable in the wild-type transcriptome, but was present at much lower levels than in degradosome- or GRSF1-impaired cells (Fig. 3b, d). To confirm the RNA-seq data, we performed a northern blot analysis using oligonucleotide probes that were complementary to the tRNA-like, its counterpart in ND4 mRNA (Fig. 3e), as well as those that were complementary to regions upstream and downstream of the

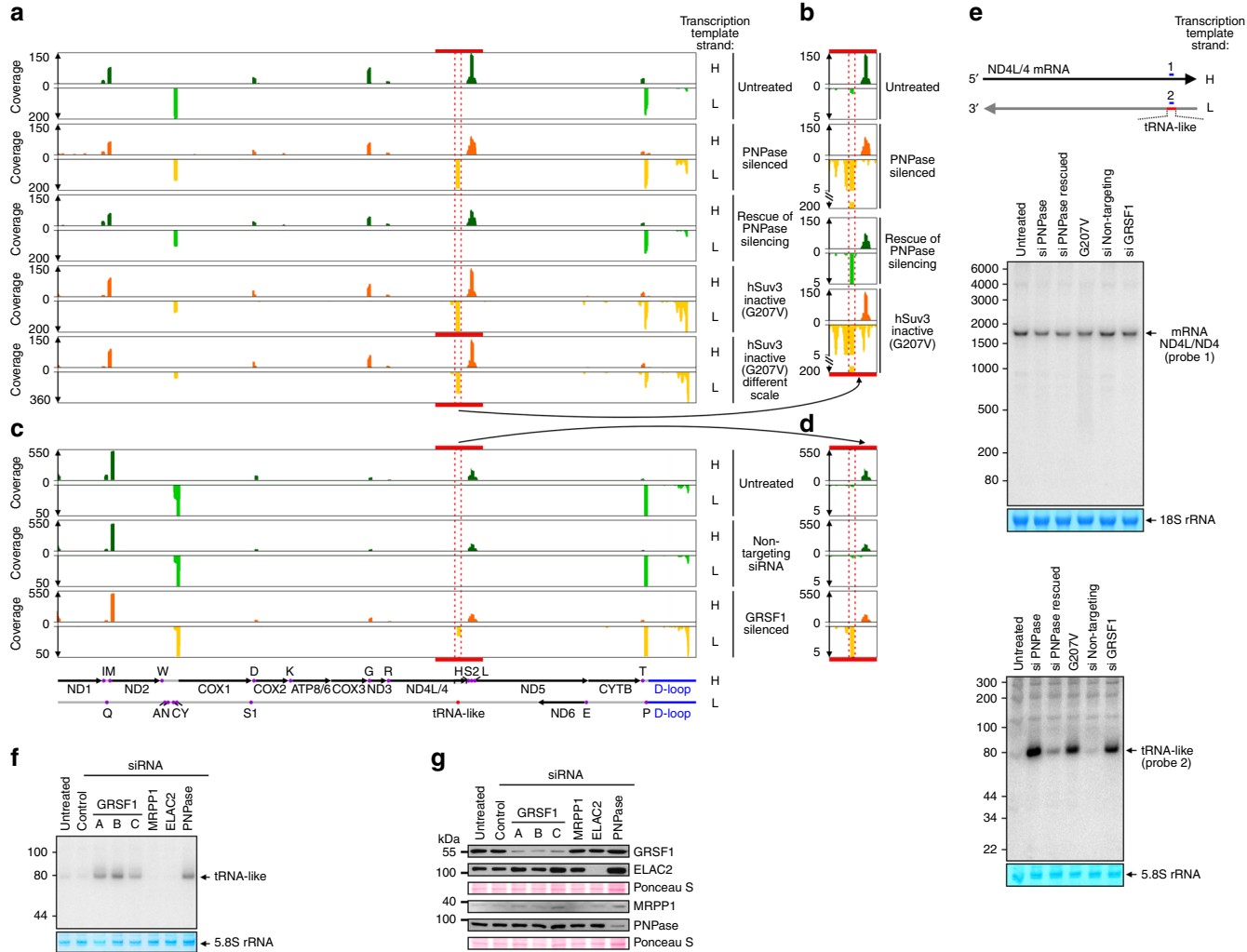

**Fig. 3** Silencing of GRSF1 or inactivation of the degradosome results in tRNA-like accumulation. **a–d** Tracks of mapped reads of short RNA libraries. The same RNA as that for Fig. 2a–d, f, g was analyzed. Position of newly identified tRNA-like RNA species is indicated by red dots. Panels **b** and **d** show the tRNA-like encoding region with a modified scale to visualize this RNA in control cells. S1 and S2 correspond to tRNA-Ser(UCN) and tRNA-Ser(AGY), respectively. **e** Confirmation of short RNA-seq results using northern blot hybridization. Strand-specific oligonucleotide probes were applied. A diagram of the probes is shown at the top. Methylene blue staining of rRNA is shown as a loading control. **f** Northern blot analysis of tRNA-like levels in untreated or siRNA-transfected cells. Three different siRNAs specific for GRSF1 were used. Methylene blue staining of rRNA is shown as a loading control. **g** Silencing of the indicated genes was confirmed by western blot analysis. Ponceau S staining of the western blot membranes is shown as a loading control. See also Supplementary Fig. 4

predicted ends of tRNA-like (Supplementary Fig. 4a). We found accumulation of the short transcript in samples originating from GRSF1- and degradosome-impaired cells (Fig. 3e). As expected, the transcript was detected only by a probe complementary to tRNA-like but not with other probes (Fig. 3e, Supplementary Fig. 4a). Notably, northern blot analysis indicated that tRNA-like accumulates in GRSF1-depleted cells to the level comparable to degradosome-deficient cells, which was not observed in RNA-seq experiment, indicating that these methods can lead to results that differ quantitatively. Nevertheless, these results confirmed the existence of tRNA-like and its accumulation after hSuv3, PNPase, or GRSF1 inactivation.

Because GRSF1 was reported to be involved in ELAC2- and RNaseP-catalyzed processing of primary polycistronic transcripts[28], we checked whether changes in tRNA-like levels observed after GRSF1 silencing are associated with impairment of this process. siRNA-mediated silencing of the expression of GRSF1, but not of the core mitochondrial processing enzymes RNAseP (subunit MRPP1) and ELAC2, resulted in accumulation

of tRNA-like (Fig. 3f, g). This effect was observed for three different GRSF1-targeting siRNAs, which indicated that the observed tRNA-like accumulation resulted from depletion of GRSF1 and was not an off-target effect of siRNA. Importantly, we confirmed that MRPP1 or ELAC2 silencing was sufficient to affect processing of mtRNA precursors (Supplementary Fig. 4b).

Altogether, these results suggested that tRNA-like is an in vivo substrate for GRSF1 and the degradosome. Moreover, the finding that depletion of essential mitochondrial processing enzymes did not upregulate tRNA-like levels indicated that the role of GRSF1 in the fate of tRNA-like is unrelated to the function of this protein in processing of primary transcripts.

Notably, in addition to tRNA-like, we observed accumulation of other short RNAs. These RNAs originate from transcription of the L-strand within the D-loop (Fig. 3a, c). The strongest effect was observed when an inactive form of hSuv3 was expressed (Fig. 3a, c), as in the case of the long RNAs (Fig. 2a–d). Nevertheless marked upregulation of D-loop originating short RNAs was also observed when PNPase or GRSF1 was silenced.

This outcome indicated another pool of RNAs which are targeted both by GRSF1 and the hSuv3–PNPase complex.

**Degradosome and GRSF1 target G-quadruplex forming RNAs.** GRSF1 is known to have a preference for binding G-rich RNAs[26]. Such G-rich sequences have the potential to form G-quadruplex structures (G4s), which suggested that GRSF1 could be required for degradation of G4-containing RNAs. Our search for G4 consensus in mitochondrial genome by means of regular expression revealed multiple sequences that had G4 forming potential (Fig. 4a). Notably, G4s were identified only in RNAs originating from L-strand transcription (Fig. 4a). This finding is in agreement with a previous report that not only developed another prediction algorithm but also performed biophysical and biochemical studies on DNA fragments derived from human mtDNA to confirm their ability to form G4s[11] (Fig. 4a). Importantly, studies of equivalent RNA and DNA sequences showed that RNA versions of G4 DNA could also form G4s. Moreover, RNA G4s are generally more stable than their DNA counterparts[35,36]. Thus, we concluded that experimentally confirmed mtDNA G4s[11] are also sequences that are most likely able to form G4s in RNA and we thus used these sequences in further analyses.

We analyzed if G4-containing RNAs are differently affected by degradosome inactivation or GRSF1 silencing compared to those RNAs that do not form G4s. To this end we calculated the coverage density of RNA-seq reads that encompass G4 sequences (Fig. 4b) or their complementary counterparts (Fig. 4c). The profile of the coverage was unaffected in the case of RNAs that did not form G4s (Fig. 4c), but was clearly altered for G4-containing RNAs (Fig. 4b). Importantly, we found accumulation of RNAs upstream of G4s (Fig. 4b), which indicated that in the presence of degradosome or GRSF1 dysfunction, transcript degradation cannot proceed efficiently across the G4 sequence. This result is in agreement with biochemical data on the degradosome components, which act from the 3′ to the 5′ end of RNA[19,20,37].

We found several sequences that strongly accumulate in GRSF1- and degradosome-deficient cells. One of stronger candidates, named Mito3, encompasses the conserved sequence block II (CSBII) located in the D-loop region. This sequence was experimentally demonstrated to form G4[13]. Examination of another accumulating transcript the tRNA-like, which we described above, showed three pairs of adjacent Gs and a G triplet—a possible G4 forming sequence.

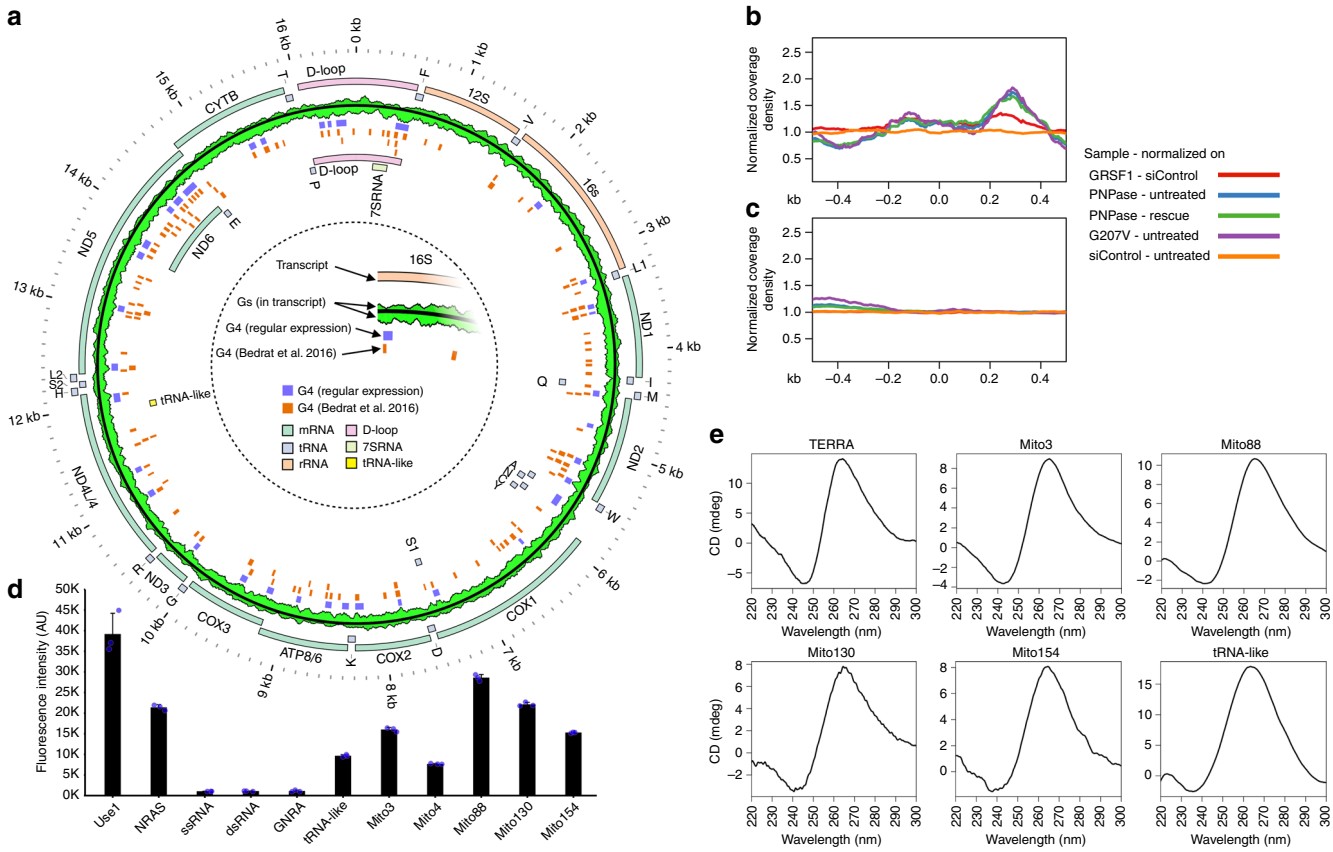

**Fig. 4** G4 RNAs, which are characteristic of L-strand transcription, are upregulated in GRSF1- or degradosome-deficient cells. **a** Localization of G4 sequences. G4s which we found by regular expression or were identified by Bedrat et al.[11] are marked. The regular expression ([gG]{3,}/w{1,30}){3,} [gG]{3,} sought a conservative consensus (4 stretches of at least 3 Gs spaced by up to 30 nucleotides). Distribution of G nucleotides across transcripts is represented by a green track. Fractions of G nucleotides were calculated within a 50 nt window slid along the genome in 5 nt steps. Features of transcripts arising from the transcription of H-strand and L-strand of mtDNA are marked as outer and inner tracks, respectively. **b**, **c** Density coverage of RNA-seq reads around G4 sequences and their complementary sequences, respectively. The horizontal x-axis was centered with respect to the center of the G4 sequence. Data from treated cells were normalized to control samples; thus, the profiles show divergence from the control. **d** ThT assay of indicated oligoribonucleotides. Well-known G4-forming oligoribonucleotides (i.e., Use1, NRAS, Mito3) as well as negative controls (i.e., ssRNA, dsRNA, GNRA) were included. Bars represent average values from three independent replicates. Error bars represent standard deviation. Blue dots represent individual values. **e** CD (220–300 nm) spectra of indicated oligoribonucleotides. See also Supplementary Fig. 5A

To confirm that the identified upregulated RNA sequences indeed form G4 structures, we performed several biochemical and biophysical tests on RNA oligos corresponding to sequences of interest (Supplementary Table 1). We first used thioflavin T (ThT), a dye that fluoresces upon contact with G4 structures present in RNA or DNA[38] and can recognize non-canonical RNA G4s[39]. We observed the expected signal increase in ThT fluorescence for two known RNAs forming G4s, as well as signals that were above that for negative controls (ssRNA, dsRNA, GNRA) for our targets (Fig. 4d). Importantly, we also noted increased fluorescence for the tRNA-like molecule (Fig. 4d). We then selected a set of five RNAs that had the strongest signal (this set included Mito3 and tRNA-like) for further investigation and performed circular dichroism (CD) measurements on the selected species. The collected spectra were indicative of G4 parallel topology (Fig. 4e). This result was expected, as in contrast to DNA G4s, the RNA G4s were previously found to universally adopt a parallel conformation[35,36]. Absorbance spectra collected over a range of temperatures allowed us to calculate thermal difference spectra (TDS), and monitor changes in absorbance at 260 and 295 nm wavelengths within a temperature gradient (Supplementary Fig. 5a). These results were also in accordance with the presence of G4 structures[40] in all tested RNAs. The exception is tRNA-like, which seems to be able to adopt various structures as revealed upon thermal melting. Taken together, our data indicate that RNAs targeted by GRSF1 and the degradosome can adopt G-quadruplex structures.

**GRSF1 melts G4 RNAs.** GRSF1 harbors three qRRM domains. Such domains were previously shown to be involved in binding and unwinding G4 structures[29]. Our analysis of available structural data on GRSF1 RNA binding domains indicated that the protein interacts with RNA in a manner, which is incompatible with G4, suggesting disruption of this structure upon GRSF1 binding (Supplementary Fig. 6). To address this possibility experimentally we used the well-characterized TERRA sequence labeled with a fluorophore (fluorescein) and quencher (dabcyl)[40]. When the G4 structure is folded, both labels are in proximity and fluorescence is quenched. Melting of the G4 structure is accompanied by an increase in the fluorescence signal due to separation of the labels[40], which was confirmed by thermal denaturation (Supplementary Fig. 5b). Importantly, upon addition of increasing amounts of GRSF1, a significant increase in fluorescence was observed (Fig. 5a) which was attributed to melting of the substrate since GRSF1 itself had no effect on fluorescence emission by fluorescein (Supplementary Fig. 5c). Moreover, the signal varied depending on the ions present in the buffer (Li$^+$ or K$^+$) which differentially affect G4 structures (Fig. 5b). Lithium ions are known to have a destabilizing effect on G4s, whereas potassium ions stabilize them[40]. Altogether obtained results suggested that GRSF1 can unwind G4. To support this conclusion further we designed a GRSF1 mutant that carries three point mutations in each of the qRRMs, which, based on structural and biochemical data of another qRRM-containing protein, hnRNP F[29], is predicted to have impaired RNA binding activity. As expected, the mutated GRSF1 protein could not unwind the TERRA substrate (Fig. 5a) as it was indeed incapable of RNA binding (see below).

Next, we used a ThT dye assay to test whether GRSF1 could melt selected mitochondrial G4 forming RNAs (tRNA-like, Mito3, Mito88, and Mito154). Increasing amounts of wild type, but not mutated, GRSF1 protein showed reductions in ThT fluorescence, indicating the disruption of G4 structures (Fig. 5c, d). Because changes in ThT fluorescence may also result from other mechanisms than G4 disruption we verified results of ThT

assay by CD measurements of the effect of GRSF1 on Mito3 structure (Fig. 5e). We have observed a marked decrease in 265 nm signal that originates from folded RNA in the presence of WT GRSF1 but not the mutated version of the protein, which confirmed that GRSF1 disrupts G4.

To unequivocally test whether the lack of the effect of the GRSF1 mutant on G4 unwinding was due to a decrease in affinity, we performed electrophoretic mobility shift assay (EMSA) experiments with tRNA-like and Mito3 substrates (Fig. 5f, top and bottom panel). The mutated form of GRSF1 had profoundly reduced affinity toward both tested RNA species. Subsequently, we tested a tRNA-like substrate in which four Gs were mutated to Cs to disrupt G-tracts. GRSF1 had significantly reduced affinity for mutated tRNA-like (Fig. 5f, middle panel).

These results indicated that GRSF1 can bind and unwind G4s via a mechanism that is dependent on the qRRM domains. Moreover, G-tracts are required for GRSF1 to efficiently bind tRNA-like.

**GRSF1 enhances degradosome-mediated degradation of G4 RNAs.** To confirm the functional link between GRSF1 and the PNPase–hSuv3 complex, we used purified proteins (Supplementary Fig. 7) for in vitro reconstitution assays to directly test whether the degradosome activity is dependent on GRSF1. As substrates we used tRNA-like and Mito3 (Fig. 6a). Notably, the PNPase or the degradosome alone could degrade tRNA-like, although at a much lower rate compared to assays in which GRSF1 was also present (Fig. 6a). This outcome was clearly visible when the reactions were compared after 15 min of incubation wherein 70% of tRNA-like was degraded by the combined action of GRSF1 and the degradosome. In contrast, PNPase or the degradosome alone degraded only 37% and 43% of the substrate, respectively, after a 15 min incubation. The observed increase in RNA degradation rate was not due to the ribonucleolytic activity associated with GRSF1, as GRSF1 alone showed no RNA degradation capacity (Fig. 6c). The GRSF1 stimulatory effect was also clear for the Mito3 substrate, which was 90% degraded after incubation with the GRSF1, hSuv3, and PNPase mixture for 15 min (Fig. 6a). In contrast, 65% and 39% of Mito3 remained intact when incubated in the presence of only PNPase or the degradosome, respectively (Fig. 6a). The stimulatory effect of GRSF1 on degradosome-mediated degradation of Mito3 was even more evident when we analyzed reaction products collected at shorter time points (Supplementary Fig. 8).

Subsequently, we studied if cooperation between GRSF1 and the degradosome requires RNA binding ability of GRSF1. To address this issue we used the mutated form of GRSF1 that we confirmed was deficient in RNA binding (Fig. 5f). We found that this form of GRSF1 did not enhance degradosome RNase activity towards tested substrates as the degradation rates seen for the degradosome alone and in the presence of mutated GRSF1 was the same (Fig. 6a). Moreover, we found that PNPase–hSuv3 mediated degradation of mutated tRNA-like, which is very weakly bound by GRSF1 (Fig. 5f), occurs at similar rates in the presence and absence of GRSF1 (Fig. 6b).

To test if the effect of GRSF1 on degradosome-mediated decay is specific for G4-forming RNAs, we examined if degradation of mutated, G4 incapable Mito3 substrates depends on GRSF1. We found that mutated versions of Mito3 in which nucleotides substitutions prevented formation of G4 (Supplementary Fig. 9a) were degraded by the degradosome at the same rate regardless the presence of GRSF1 (Supplementary Fig. 9b).

Finally, we repeated the degradation assays including PNPase with GRSF1 to assess whether GRSF1 itself can unwind the G4 substrates for PNPase. The obtained results indicated that

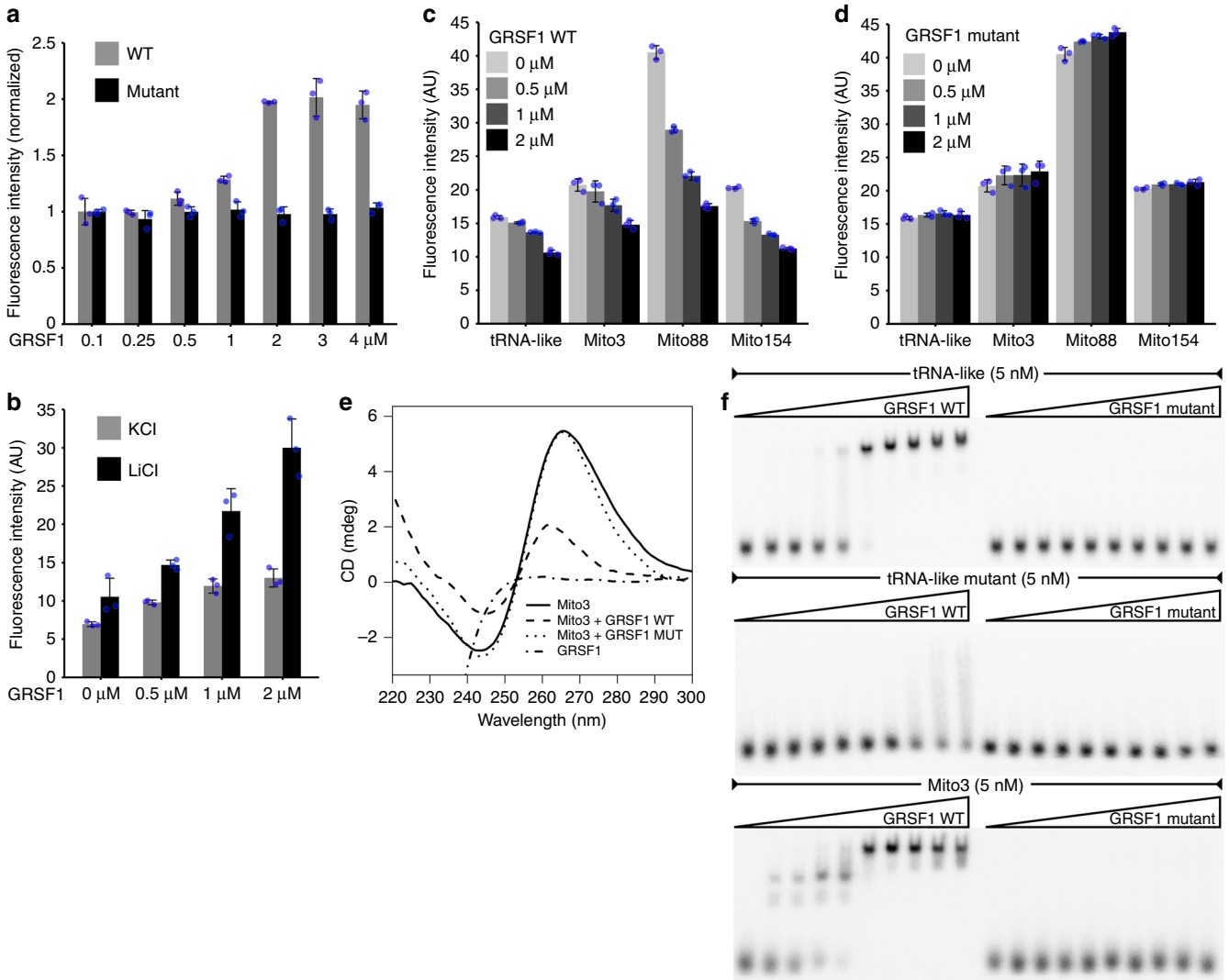

**Fig. 5** GRSF1 binds and melts G4 RNAs in vitro, which is dependent on qRRM domains. **a** Analysis of the ability of wild type or mutated GRSF1 to melt G4 structures. Mutated GRSF1 harbors point mutations in qRRM domains. Fluorophore and quencher assays were performed. **b** Confirmation that wild-type GRSF1 targets G4 structures. Different ionic conditions that are known to affect G4 structure stability were applied. The presence of potassium ions stabilizes G4 whereas the structures are less stable in the presence of lithium ions. **c, d** ThT in vitro assay of mitochondrial G4 RNAs, which are upregulated in GRSF1- or degradosome-deficient cells. Wild type (**c**) or mutated (**d**) GRSF1 was examined. Bars represent average values from three independent replicates. Error bars and blue dots represent standard deviation and individual values, respectively (**a–d**). **e** CD (220–300 nm) spectra of Mito3 RNA, GRSF1, or mixture of thereof. **f** EMSA assays of indicated substrates. The tRNA-like mutant has mutated G-tracts. See also Supplementary Fig. 5b, c and Supplementary Fig. 7

GRSF1 is not sufficient to prepare G4 substrates for PNPase as substrate degradation by the PNPase and GRSF1 mixture was inefficient and was in fact lower than that for PNPase alone (Fig. 6c). Similar results were obtained when non-hydrolyzable ATP analogs were used, which points to importance of not only hSuv3 presence but also its ability to express ATP-dependent activity (Supplementary Fig. 10).

As such, this result highlights the requirement of the degradosome complex for efficient degradation of RNAs as well as the importance of the hSuv3 helicase. Moreover, our results demonstrated that GRSF1 stimulates the ribonucleolytic capacity of the hSuv3–PNPase complex towards G4-containing substrates in RNA binding-dependent manner.

**GRSF1 was acquired by mitochondria to promote G4 RNA decay.** GRSF1 belongs to the qRRM protein family, which seems to be absent in bacteria from which mitochondria originated.

Therefore, we investigated when in the process of evolution GRSF1 was acquired, and whether among qRRM proteins the mitochondrial localization of GRSF1 is conserved and specific. We performed analysis using BlastP with human GRSF1 sequence and GRSF1 consensus made out of three qRRM domains as search queries. We found that GRSF1 is present in Deuterostomia and further examination revealed that it is exclusively present in vertebrates (Fig. 7a). The vertebrate organisms for which the GRSF1 was not found either did not have available genome sequences (i.e. Dipnoi, Cladistia, Chondrostei) or the genes were not assigned to the available sequence data. For the latter cases (i.e. Cyclostomata and Elasmobranchii), we performed BlastN on whole genome contigs. This analysis revealed the presence of GRSF1 genes in these organisms, thus corroborating the presence of GRSF1 in vertebrates. Together these results indicate that GRSF1 is unique to vertebrates (Fig. 7a).

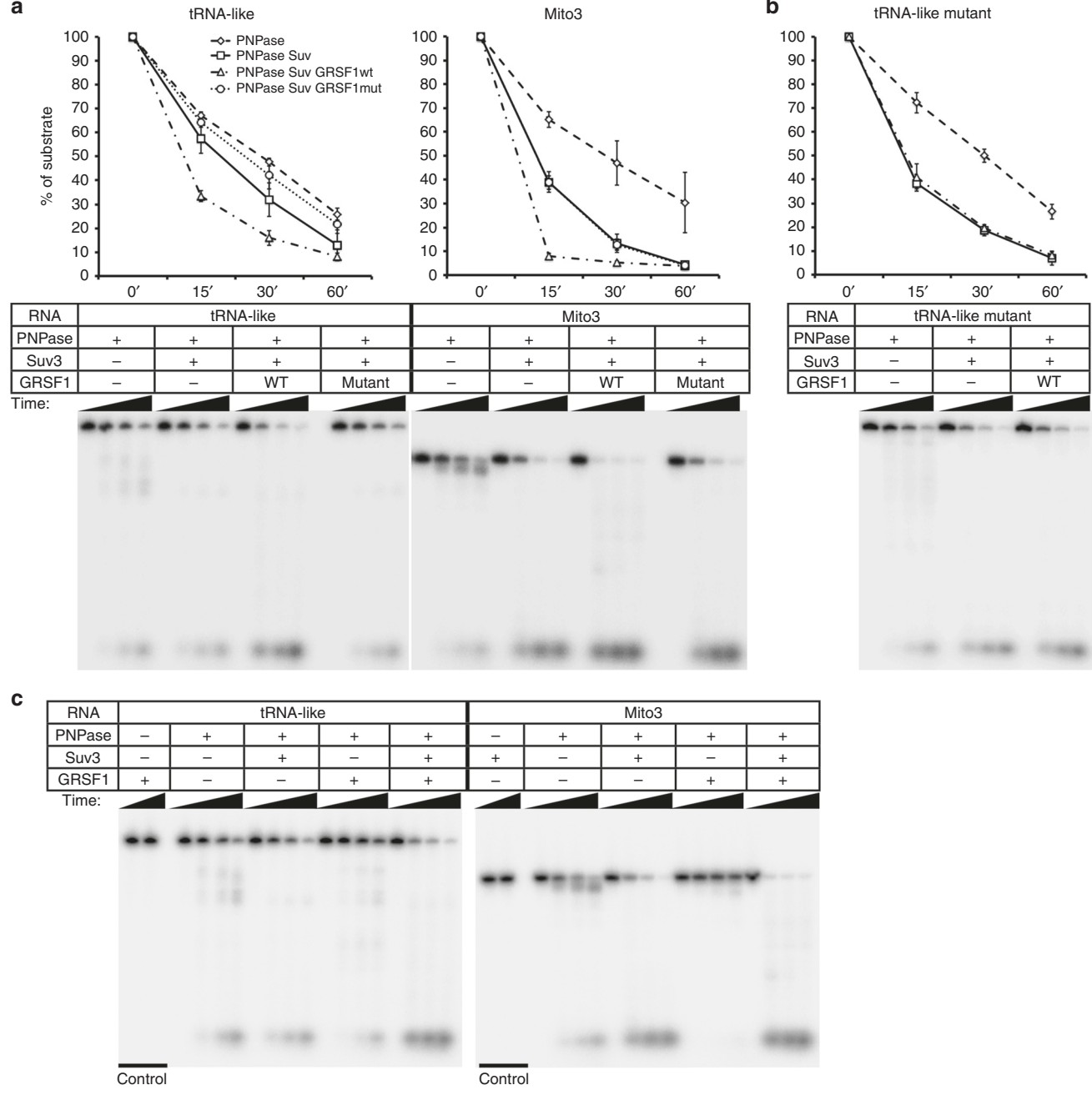

**Fig. 6** GRSF1 augments degradosome-mediated degradation of G4 RNAs. **a–c** In vitro degradation of indicated substrates by recombinant PNPase, hSuv3, GRSF1 or their combined action. Graphs show quantified substrate decay. Following time points were analyzed: 0, 15, 30, and 60 min. Data points represent the mean for three independent experiments. Error bars represent standard deviation. See also Supplementary Figs. 7–10

To compare GRSF1 with other qRRM proteins, we used sequences of all qRRM proteins from 10 GRSF1-positive and 3 GRSF1-negative organisms (Supplementary Table 2) and performed multiple sequence alignments using the T-Coffee server[41] to generate a phylogenetic tree[42]. This analysis showed that the GRSF1 proteins are clustered in one branch, which suggests that they constitute a distinct group among qRRM proteins (Fig. 7b). We then took all identified qRRM protein sequences and performed in silico prediction of their cellular localization using three independent tools[43–45]. All the GRSF1s were predicted to localize to mitochondria by at least two algorithms, and in most cases (9 out of 11) by all three (Fig. 7c). In contrast, the non-GRSF1 qRRM proteins were not predicted to localize to

mitochondria, or in the minority of cases (6 out of 79) were predicted by just one tool.

We then investigated if the mitochondrial genomes of the GRSF1-positive organisms differed from that of GRSF1-negative organisms. Analysis of 3932 mitochondrial genomes revealed that GRSF1-positive organisms have significantly lower AT content, negative GC skew, and increased number of G4s (Fig. 7d). To further test the latter parameter, we used QGRS mapper[46] as well as regular expression with a more conservative consensus (4 stretches of at least 3 Gs spaced by up to 30 nucleotides) to examine 13 organisms containing the qRRMs we analyzed above. Both methods showed that G4 consensus sequences are over-represented by an order of magnitude in mitochondrial genomes

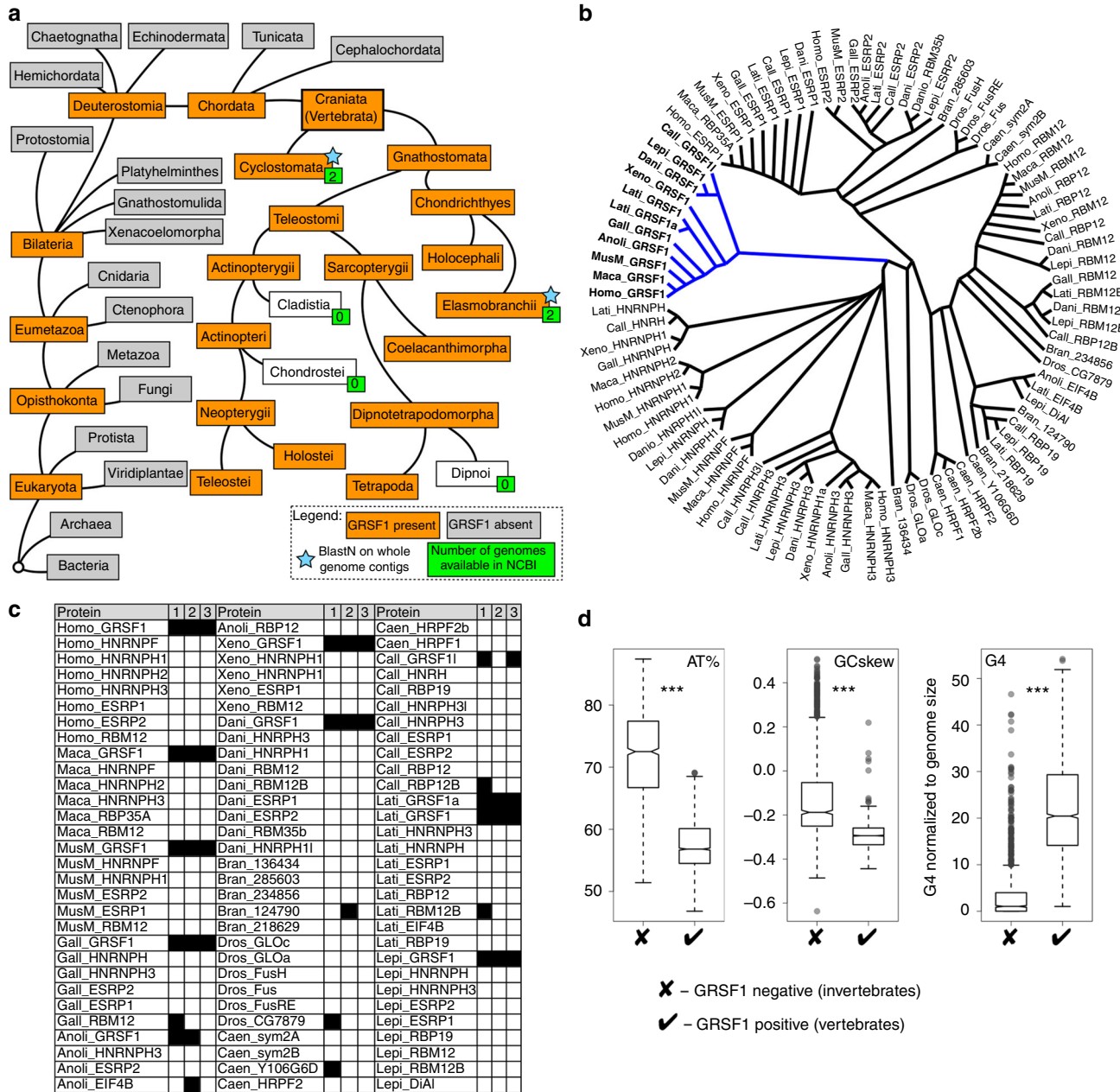

**Fig. 7** GRSF1 proteins form a distinct group of qRRM proteins that may have been acquired by mitochondria in response to changes in nucleotide composition of mitochondrial genomes. **a** Branching tree diagram showing organisms that do or do not possess GRSF1 (NCBI taxonomy). **b** Phylogenetic tree of qRRM proteins from the following organisms: Homo—*Homo sapiens*, Maca—*Macaca mulatta*, MusM—*Mus musculus*, Gall—*Gallus gallus*, Anoli—*Anolis carolinensis*, Xeno—*Xenopus laevis*, Dani—*Danio rerio*, Call—*Callorhinchus milii*, Lati—*Latimeria chalumnae*, Lepi—*Lepisosteus oculatus*, Bran—*Branchiostoma floridae*, Dros—*Drosophila melanogaster*, Caen—*Caenorhabditis elegans*. Multiple sequence alignments were performed using the T-coffee server. The tree was rendered using Dendroscope[69]. **c** In silico prediction of localization of qRRM proteins by three independent tools: 1. TargetP, 2. MitoProt, and 3. MultiLoc2. **d** Box plots representing AT content, GC skew, and number of G4 sequences identified by regular expression normalized with respect to genome size [G4 number/(length of mitochondrial genome/mean length of mitochondrial genome)]. Metazoan mitochondrial genomes (2832 vertebrates and 1100 invertebrates) were downloaded from the MetAMiGA database[70]. A Wilcoxon rank-sum test was used for statistical analysis of data (***$P < 0.0001$). The box in the boxplot represents interquartile, while upper and lower whiskers represent maximum and minimum values, respectively (excluding outliers, which are represented by semi-transparent dots). The band represents median, while the notch 95% confidence interval of the median

of GRSF1-positive organisms (Supplementary Table 2). To test if the number of G4 consensus sequences in these mitochondrial genomes could be explained by the AT content and the GC skew, we generated 10,000 random sequences of the same length, AT content, and GC skew as the mitochondrial genome for each species tested and calculated the average number of G4 consensuses within those sequences. Interestingly, the number

of G4 sequences in real genomes was greater than that for artificially generated genomes (Supplementary Table 2). This outcome suggests that there are other factors besides AT content and GC skew that contribute to the number G4 sequences in mitochondrial genomes.

In summary, our analyses suggested that GRSF1 was acquired relatively late in evolution with the appearance of vertebrates.

This acquisition coincided with changes in mitochondrial genomes (GC skew and AT content) that resulted in a substantial increase in sequences with G4-forming potential.

## Discussion

RNA molecules can form many different structures, many of which involve non-canonical base pairing. Four-stranded structures known as G4s have been shown to be important in various aspects of RNA biology. The extraordinary GC skew of vertebrate mitochondrial genomes results in synthesis of G-rich RNAs that are prone to form G4s. Such RNAs, which are mostly antisense to functional RNAs, are transcribed at high rates but their steady-state levels are extremely low. In this study, we found that the mitochondrial degradosome associates and cooperates with GRSF1 to control G4-containing mtRNAs. We revealed a crucial mechanism that prevents accumulation of antisense transcripts in human mitochondria, which is of great importance for unperturbed expression of mitochondrial genes essential for humans.

Results from biochemical assays allowed us to propose a molecular mechanism by which GRSF1 facilitates RNA degradation by the mitochondrial degradosome. In this mechanism, GRSF1 binds G4 and melts the G4 structure, but the RNA remains tightly bound to GRSF1, which presents an obstacle for PNPase activity (Fig. 8a). Thus, hSuv3 is needed to liberate RNA from GRSF1 and to unwind dsRNA stretches (Fig. 8b). Our degradation assays suggest that the dissociation of GRSF1 from RNA by hSuv3 is more efficient than unwinding of G4 by the helicase. This is corroborated by our finding that the GRSF1-hSuv-PNPase combination had the highest G4 RNA degradation efficiency. To the best of our knowledge, this is the first experimental example of cooperation between a G4 melting protein and other molecular machinery. Furthermore, our studies identified for the first time a factor that augments the activity of the degradosome.

In contrast to activators, two antagonists of degradosome-mediated RNA decay were previously identified. The LRPPRC–SLIRP complex was suggested to regulate degradosome-dependent mtRNA turnover[22], whereas the FASTK protein was shown to control exonucleolytic, degradosome-mediated processing of the 3′ UTR of ND6 mRNA[24]. In the case of LRPPRC–SLIRP, recent study based on RNase footprinting and PAR-CLIP showed its preference for binding of mRNAs which affects their secondary structure and translation ability[47]. Analysis of LRPPRC-binding sites revealed a highly degenerate consensus recognition sequence[47]. Importantly, G residues were strongly underrepresented in this 40 nucleotide long sequence. Such results are in accord with our findings, which indicate that GRSF1-augmented degradosome-mediated decay has specificity for G-rich regions. Thus, we can speculate that G-poor sequences, which are enriched in mRNAs are protected from degradation by LRPPRC-SLIRP complex while G-rich non-coding RNA species are degraded with the help of GRSF1.

RNA-seq analysis of degradosome-impaired cells provided the first picture of the mitochondrial transcriptome in the presence of RNA decay inhibition. The extent to which non-coding RNAs accumulate, especially in hSuv3-G207V-expressing cells, implied that the hSuv3–PNPase complex plays a primary role in regulating non-coding mtRNAs, which mostly are produced by L-strand transcription. The importance of PNPase- and Suv3-dependent processes is underscored by the embryonic lethality of mice with knockouts of these genes[48,49] as well as by an increasing number of identified pathogenic mutations in humans[50,51]. Indeed, Suomalainen and co-workers directly linked Leigh syndrome to impairments in degradosome-dependent mtRNA degradation[51], thus confirming the physiological significance of this process. Results of our studies can be of great value in complete understanding of diseases' mechanisms. Interestingly, in bacteria, degradosome activity involves an endoribonuclease that provides an entry site for exonucleolytic activity. Whether this situation also occurs in human mitochondria remains to be established. If so, initial endonucleolytic cleavage of long L-strand transcripts could improve their degradation by the hSuv3–PNPase. Two candidates for such a mitochondrial endoribonuclease are of particular interest: LACTB2[52] and YBEY[53]. Their functional interplay with the hSuv3–PNPase complex awaits investigation.

Analysis of short RNAs that accumulate upon GRSF1 or degradosome dysfunction showed strong accumulation of the tRNA-like. Whether this transcript plays any role in mitochondrial function or is merely a by-product of the processing of primary RNA requires further studies. When the activity of nuclear RNA surveillance pathways is altered, a significant number of transcripts, which are not detected upon normal conditions, accumulate[54]. tRNA-like is post-transcriptionally modified by the addition of non-templated CCA at the 3′ end. This modification was already observed by others for

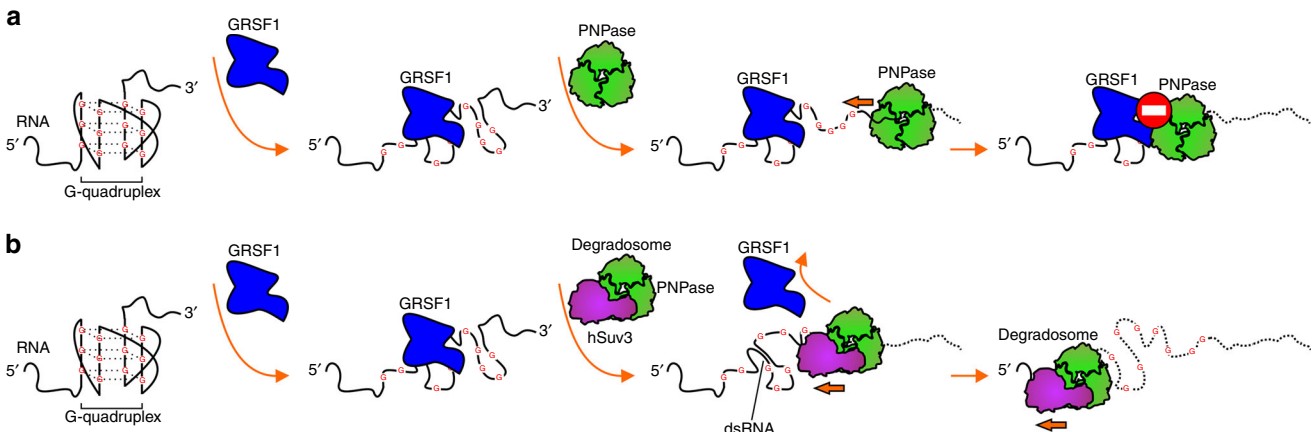

**Fig. 8** Interplay between PNPase, hSuv3, and GRSF1. Two scenarios are considered—without and with hSuv3 (**a** and **b**, respectively). **a** Unreal conditions when hSuv3 is not present. PNPase degrades single-stranded RNA without involvement of other proteins. GRSF1 binds and melts G4 structures but forms an obstacle for PNPase. **b** Actual conditions when hSuv3 helps to dissociate GRSF1 from RNA and unwinds double-stranded RNA structures, which enables complete degradation of the substrate by PNPase. Lack of GRSF1 hampers degradation of G4 structures by the degradosome

mitochondrial non-tRNA molecules[55]. Importantly, one of the previously identified sites of addition of non-templated CCA[55] is consistent with the 3′ end of the tRNA-like. The function of this modification for non-tRNA species is currently unclear. Nuclear-encoded, structurally unstable tRNAs and tRNA-like species undergo quality control that involves the addition of CCACCA sequences[56]. However, this type of modification was not observed either by us or others[55] in mitochondria.

The data presented here have some interesting evolutionary implications since we show that GRSF1 is the only qRRM protein present in mitochondria. We propose that changes to the mitochondrial genome resulted in the increased presence of G4 sequences and that acquisition of GRSF1 was a response to this change that allowed maintenance of most G4 RNAs in an unwound state. Guo and Bartel[57] recently showed that the majority of predicted G4s are in fact unfolded in the cell[57]. Based on this finding, they propose that either an organism does not have G4 sequences (e.g., prokaryotes) or it developed counter-measures to preserve the majority of these G4 RNAs in an unfolded state (eukaryotes). The acquisition of G4 sequences seems to correlate with the evolution of biological complexity. Our work showed a similar phenomenon in mitochondrial genomes. Only the mitochondrial genomes from complex life forms underwent a transition from G4-poor to G4-rich. Our data suggest that the evolutionary response to the increasing number of G4s was acquisition of a protein (GRSF1) that can melt such structures.

The question remains: why did G4 sequences appear in the first place? Expression of G4 sequences in bacteria impairs their growth and produces multiple defects such as perturbed translation, increased stop codon read-through and frame shifting[57]. This finding, together with the fact that G4 structures are under surveillance of cellular machinery, suggests that G4s pose a problem for the cell rather than an advantage. Indeed, the presence of G4 structures in mtDNA was linked with the occurrence of deletions[58]. Guo and Bartel proposed that maintenance of G4 melting machinery in prokaryotes is costly, and the minimal genome would decrease the likelihood of the appearance of G4s. In contrast, in eukaryotes the presence of G4s would provide opportunities for regulation. Which scenario would apply to mitochondrial genomes is currently unclear. It was shown that GTPase activity of NOA1, which functions in mitochondrial ribosome assembly, is influenced in vitro by G4 RNAs[59]. Moreover, TFAM, a major mitochondrial nucleoid protein that is essential for transcription and replication, was shown to have versatile G4 binding ability[60]. The exact role of these features is unknown, but it seems likely that G4s RNA can influence the activity of mitochondrial proteins. Thus, not surprisingly dysfunctions of the degradosome components or GRSF1 are manifested by pleiotropic mitochondrial phenotypes including abnormal mitochondrial translation, affected steady-state levels of functional RNAs, and impeded maintenance of mitochondrial genome[15,27,28,61].

## Methods

**Cell lines and cell culture.** Immunofluorescence experiments were performed using HeLa Flp-In T-Rex (a kind gift from Matthias Hentze). Other experiments were performed using 293 Flp-In T-REx cells (R78007; Thermo Fisher Scientific) or their stably transfected derivatives. 293 Flp-In T-REx cells expressing hSuv3-TAP, PNP-TAP, mt-TAP, or a catalytically inactive version of hSuv3 (G207V) were described previously[16]. 293 Flp-In T-REx cells expressing an siRNA-insensitive form of PNPase were described elsewhere[21]. Cells were cultured at 37 °C under 5% $CO_2$ in Dulbecco's modified Eagle's medium (DMEM) medium (Gibco) supplemented with 10% fetal bovine serum (Gibco). Transgenes were induced with tetracycline (25 ng/ml). The identity of parental 293 cells and HeLa cells was confirmed using STR profiling by DSMZ (Germany). Cells were tested for mycoplasma contamination.

**siRNA transfection.** Cells were transfected using RNAiMAX (Thermo Fisher Scientific) in the forward manner according to the manufacturer's instructions and collected after 3 days. The final concentration of siRNA was 20 nM. All siRNAs were purchased from Invitrogen (Table 1).

**TAP-tag purification and mass spectrometry.** Three independent replicates were carried out for all purifications. Mitochondria isolation and single-step TAP-tag purification (without chromatography on calmodulin beads) were performed as described previously[16]. Eluted proteins were identified using MS/MS analysis. MS analysis was performed by LC-MS in the Laboratory of Mass Spectrometry (IBB PAS, Warsaw). Samples were processed and subjected to label-free quantification as described elsewhere[62]. Protein abundance was defined as the mean signal intensity calculated by MaxQuant software for a protein divided by its molecular weight. Enrichment was defined as the ratio of mean protein intensity measured in three bait purifications relative to background level (i.e., mean protein intensity in three mt-TAP purifications). Putative degradosome-associated proteins met the following criteria: (1) were present in all purifications of hSuv3-TAP and PNP-TAP, (2) were enriched by at least three-fold for both baits, (3) were previously suggested to be mitochondrial proteins (a list is provided in the sheet "Mitochondrial proteins", in Supplementary Data 1).

**RNA isolation and northern blot.** Total RNA was isolated with the TRI Reagent according to the manufacturer's instructions. Standard northern blotting (Fig. 3e-probe 1 and Supplementary Fig. 4b) was performed as described previously[16]. All other northern blots were performed as follows: 3 µg of total RNA was mixed with an equal volume of denaturing loading dye (95% formamide, 20 mM EDTA, 0.25% bromophenol blue, 0.25% xylene cyanol), heat denatured for 3 min at 85 °C, run on a 10% polyacrylamide/8 M urea gel in 1× TBE using a Mini-Protean II apparatus (BioRad) and blotted onto Hybond-NX membranes (GE Healthcare) by wet electrotransfer in 0.5× TBE buffer (14–16 h at constant amperage of 20 mA) using a Mini Trans-Blot Cell apparatus (BioRad). After transfer of nucleic acids, membranes were stained with methylene blue solution (0.02% methylene blue in 0.3 M sodium acetate, pH 5.2). All hybridizations were performed overnight in PerfectHyb Plus buffer (Sigma) at 37 °C. The following strand-specific PNK-radiolabeled oligoprobes were used: RSZ654 GTAGGA-GAGTGATATTTGATCAGGAGAACGT (ND4L/ND4, Fig. 3e, probe1), RSZ655 ACGTTCTCCTGATCAAATATCACTCTCCTAC (tRNA-like, Fig. 3e, probe 2, Fig. 3f), RSZ950 TATACTCCCTCTACATATTTACCACAACAC (upstream of tRNA-like, Supplementary Fig. 4a), RSZ949 CCCACTATTAACCTACTGGGAGA ACTCTCT (downstream of tRNA-like, Supplementary Fig. 4a), RSZ129 GCGGTCAAGTTAAGTTGAAATCTCCTAAGTGTAAGTTGGGTGCT TTGTGT (tRNA Val, Supplementary Fig. 4b). Membranes were washed three times with 2× SSC solution for 10 min at 37 °C. After overnight exposure to PhosphorImager screens (FujiFilm), autoradiograms were processed using a Typhoon FLA 9000 scanner (GE Healthcare). Data were analyzed by Multi Gauge V3.0 software (FujiFilm).

**Western blot.** Total protein cell extracts were prepared in lysis solution (10 mM Tris, 140 mM NaCl, 5 mM EDTA, 1% (v/v) Triton X-100, 1% (w/v) deoxycholate,

---

### Table 1 siRNAs used in the study

| Targeted gene | Name in the manuscript | Producer ID | Entrez Gene ID |
|---|---|---|---|
| PNPT1 | PNPase | PNPT1HSS131758 | 87178 |
| TRMT10C | MRPP1 | RG9MTD1HSS123552 | 54931 |
| ELAC2 | ELAC2 | ELAC2HSS127087 | 60528 |
| GRSF1 | A | GRSF1HSS104516 | 2926 |
| GRSF1 | B (used in RNA-seq experiment) | GRSF1HSS104517 | 2926 |
| GRSF1 | C | GRSF1HSS179021 | 2926 |
| Stealth RNAi siRNA negative control Hi GC | Non-targeting or Control | 12935400, 46-2000 | Not applicable |

0.1% (w/v) SDS). Protein concentration was determined by the Bradford method. Protein extracts (30 μg per lane) were separated by SDS-PAGE and transferred to a nitrocellulose membrane (Protran, Whatman GmbH). Western blotting was performed according to standard protocols using the following primary antibodies: anti-FLAG (dilution 1:10,000, PA1-984B; Thermo Fisher Scientific), anti-PNPase (dilution 1:500, sc-49315; Santa Cruz Biotechnology), anti-GRSF1 (dilution 1:1,000, a kind gift of Jeffrey Wilusz[26]), anti-hSuv3 (dilution 1:1,000, GTX628260; Gene-Tex), anti-MRPP1 (dilution 1:250, ab94643; Abcam), and anti-ELAC2 (dilution 1:250, 10071-1-AP; Proteintech). Appropriate horseradish peroxidase-conjugated secondary antibodies (dilution 1:15,000, 401393, 401215, Calbiochem and A5420, Sigma) were detected by enhanced chemiluminescence (170-5061; BioRad) according to the manufacturer's instructions. Uncropped scans of blots are shown as Supplementary Fig. 11.

**Immunofluorescence**. HeLa cells were transiently co-transfected with the following BiFC DNA constructs[21]: (1) active degradosome—pRS409 (hSuv3WT pBiFC-VC155) and pRS411 (PNPaseWT pBiFC-VN173), (2) inactive degradosome—pRS415 (hSuv3-G207V pBiFC-VC155) and pRS505 (PNPaseR445446E_pBiFC-VN173). TransIT2020 Reagent (Mirus) was used for transfection according to the manufacturer's recommendations. At 24 h after transfection the cells were plated on glass coverslips and cultured for an additional 24 h. The cells were then washed with PBS twice and treated for 30 min with 5% (v/v) formaldehyde and 0.25% (w/v) Triton X-100 solution in PBS. Cells were washed with PBS three times and incubated for 30 min with 3% (w/v) BSA in PBS before an overnight incubation at 4 °C with the primary antibody anti-GRSF1 (HPA036985, Sigma) diluted 1:200 in 3% BSA in PBS. After washing with PBS, relevant secondary antibodies conjugated with Alexa555 were applied for 1 h at room temperature. Mitochondria were stained by addition of MitoTracker DeepRed (200 nM) to the culture medium 1 h prior to fixation. Cells were imaged with a FluoView1000 confocal microscope (Olympus) and a PLANAPO 60.0 × 1.40 oil objective. Images were collected with 215 nm resolution. Object-based colocalization was performed with Imaris 7.2.3 software. Objects with centers separated by <215 nm were considered to be colocalized.

**Library preparation and next-generation sequencing**. RNA-seq experiments were performed in triplicate. Libraries of long RNAs were prepared as described in detail elsewhere[63] except that a Ribo-Zero rRNA Removal Kit (Human/Mouse/Rat, Illumina) was used for rRNA removal. Libraries for analysis of small RNAs were prepared as described by Labno et al.[64] except that total RNA, which was not subjected to rRNA depletion, was used for library preparation. Sequencing, whose results are presented in Figs 2a and 3a, was carried out on an Illumina HiSeq 1500 sequencing platform, using TruSeq SBS Kit v3 (FC-401-3001, Illumina), TruSeq PE Cluster Kit v3-cBot-HS (PE-401-3001, Illumina) and standard libraries denaturation and pair-end sequencing procedures of 2 × 75 cycles (Instructions: 15050107 Rev. B, 15035788 Rev. E). Sequencing (Figs 2c and 3c) was carried out on an Illumina NextSeq 500 sequencing platform, using NextSeq 500 High Output Kit (150 cycles) (FC-404-1002, Illumina) and standard libraries denaturation and pair-end sequencing procedures of 2 × 75 cycles (Instructions: 15048776 Rev. D, 15046563 Rev. F).

**Quality control, trimming, and mapping of NGS reads**. Sequences of all RNA-seq reads were striped of adapters and trimmed from poor quality bases using cutadapt (version 1.9.2.dev0). Long RNA libraries were mapped using the GSNAP short read aligner (version 2016-05-01) allowing for spliced read alignment and default settings. The assemblies were prepared with reference human genome GRCh38 facilitating the alignment of known splice junctions of the basic GENE-CODE v26 annotation and treating the mitochondrial genome as a circular DNA molecule to allow for mapping of all reads in the D-loop region[65]. Reads from short RNA-seq libraries were additionally merged within pairs and only ones which overlapped by at least five bases were further aligned to the genome using GSNAP short read aligner. The quality control, read processing and filtering, visualization of the results, and counting of reads for the annotation was performed using custom scripts incorporating elements of the RSeQC (version 2.6.1), BEDtools (version 2.23.0), and SAMtools packages. The reference transcript annotation used was the GENECODE v26. To prepare figures tracks of mapped reads were displayed using the Integrative Genomics Viewer (IGV) and assembled using graphical software (CorelDraw).

**Contribution of L-strand RNAs to mitochondrial transcriptome**. Uniquely mapped reads were used to prepare pileups of expression corrected to the total number of reads mapping to both strands of the mitochondrial genome. The coverage over the non-coding bases of the mitochondrial genome, additionally excluding regions antisense to rRNA and tRNA to avoid bleed-trough from the other strand, was summarized separately for the L-strand and H-strand utilizing custom scripts incorporating elements of the RSeQC and bwtool (version 1.0) package and visualized in R.

**Calculation of cumulative coverage of G4-containing RNAs**. We used deeptools bamCoverage software to transform alignment of reads to genome coverage.

Subsequently, computeMatrix software was used to calculate average scores for designated genome regions. Results were normalized and visualized using R.

**Re-analysis of ENCODE data**. Publicly available GRSF1 eCLIP datasets with their corresponding mock controls (accession IDs: ENCSR668MJX, ENCSR638YHN) have been downloaded from the ENCODE data repository (https://www.encodeproject.org/search/?type=Experiment&ass) and mapped utilizing our pipeline as described for long RNAseq libraries.

**Identification of G4 sequences and data visualization**. The regular expression search, genome features calculation (AT%, GC skew), sliding window calculations, sequence generation, and data visualization were performed using R, Circos, Python, and Biopython. Additionally, QGRS Mapper[46] was used to identify G4 consensus sequences.

**G4 annealing procedure**. RNA oligomers in annealing buffer (12.5 mM LiCacodylate, pH 7.0, 5 mM KCl) were heated to 85 °C for 5 min and over the course of 3 h were cooled slowly to room temperature. The samples were then incubated at 4 °C overnight.

**Fluorophore and quencher assay**. TERRA RNA oligomer labeled with fluorophore and quencher (Supplementary Table 1) was added to a final concentration of 1 μM in a 20 μl reaction volume. The remaining reaction components were 20 mM LiHEPES pH 7.0, 100 mM KCl (or 100 mM LiCl) and indicated protein concentrations. Fluorescence signal measurements were performed using a Light-Cycler 480 real-time system (Roche) in a 384-well format.

**Thioflavin T assay**. Reactions containing 20 mM LiHEPES pH 7.0, 100 mM KCl, 1 μM Thioflavin T, 1 μM RNA oligo, and indicated concentrations of protein in 20 μl were measured using 384-well flat bottom transparent polystyrol plates (Greiner) and an Infinite M1000 plate reader (Tecan) with excitation and emission wavelengths of 430 nm and 490 nm, respectively, and a 10 nm bandwidth.

**Biophysical assays**. RNA oligos (1 μM) were annealed in 10 mM LiCacodylate pH 7.0 and 5 mM KCl buffer. Measurements were made using a Jasco J-810 CD spectrometer equipped with a thermostat-controlled cell holder (JASCO Corporation) and a 1 cm path-length quartz cuvette with a magnetic stirrer. Three scans at 25 °C between 220 and 300 nm were recorded, averaged, and buffer-corrected. For thermal denaturation studies, absorbance spectra and CD spectra were recorded between 220 and 320 nm, from 3 °C to 98 °C with 5 °C steps, and a 2 min equilibrium time. Data were analyzed with Spectra Manager v2 (JASCO Corporation) and R.

**Radioactive labeling of substrates**. 5′-end labeling of substrates was performed using [γ-$^{32}$P] adenosine triphosphate (Hartmann Analytic) with T4 polynucleotide kinase (NEB). After labeling, oligonucleotides were subjected to phenol–chloroform extraction, precipitated, and purified by denaturing polyacrylamide gel electrophoresis.

**Electrophoretic mobility shift assay**. EMSAs were performed using a constant RNA concentration of 5 nM and increasing protein concentrations (0, 10, 20, 40, 80, 160, 320, 640, 1280, and 2560 nM). Binding proceeded for 30 min at 30 °C in 15 μl reaction volumes containing 10 mM Tris-HCl pH 8.0, 2.5 mM KCl, 30 mM NaCl, 1 mM EDTA, and 0.2 mM DTT enriched with 75 μg/ml heparin. Samples were mixed with 5 μl Gel Loading Dye (Thermo Fisher Scientific) and separated on a 7% polyacrylamide gel (19:1) run at constant 10 W. The gels were exposed in Phosphorimager cassettes overnight and the results were recorded with a Typhoon FLA 9000 scanner (GE Healthcare). Quantification of the results was performed using ImageJ software.

**RNA degradation assays**. Degradation assays were carried out at 37 °C in a 30 μl reaction volume containing 1 pmol RNA and 3 pmol of the indicated protein/s in 20 mM Tris-HCl pH 8.0, 50 mM NaCl, 10 mM $NaH_2PO_4$, 2 mM $MgCl_2$, 1 mM ATP, 1 mM DTT. Reactions were started by the addition of annealed RNA enriched with ATP. Measurements were collected at 0, 15, 30, and 60 min. For the control samples (without protein), measurements were taken at 0 and 60 min. Reactions were stopped at the indicated time points by addition of an equal volume of RNA loading dye (98% deionized formamide, 25 mM EDTA pH 8.0, 0.01% (w/v) xylene cyanol, and 0.01% (w/v) bromophenol blue) and then flash-frozen in liquid nitrogen. When all samples were collected, they were heated for 7 min at 85 °C. Subsequently, 10 μg proteinase K was added to each sample and they were incubated for 10 min in RT. The samples were then separated on a denaturing gel (8 M urea, 20% acrylamide in 1× TBE) run at constant 20 W. The gels were exposed in Phosphorimager cassettes overnight and the results were recorded with a Typhoon FLA 9000 scanner (GE Healthcare). Quantification of the results was performed using ImageJ software.

**Sequence alignment, dendrogram, and localization prediction**. We used three localization prediction servers, each using a different algorithm: TargetP[44], MultiLoc2[45], and MitoProt[43]. Protein sequence alignments were compiled using the T-coffee server[41] and Simple Phylogeny (EMBL-EBI) with neighbor-joining clustering[42].

**Protein purification**. hSuv3 (47-786 aa) and GRSF1 (118-480 aa, NP_002083.3) were expressed as N-terminal 6xHis-SUMO-tagged proteins whereas PNPase (38-783 aa) was expressed as C-terminal 6xHis fusion protein. The truncation of GRSF1 was designed based on prediction of the mitochondrial targeting sequence using MitoProt II v1.101. The point mutations in GRSF1 (Q155A, W159A, E223A, R255A, Y259A, E320A, R406A, F410A, and E470A) were designed based on structural studies of the qRRM domain of the hnRNP F protein[29]. hSuv3 and PNPase were truncated at the N-terminus in agreement with previous reports[66,67]. For proteins overproduction *Escherichia coli* BL21 strain was transformed with appropriate plasmids. Bacteria were cultured in autoinduction Super Broth base including trace elements medium (Formedium), supplemented with 2% glycerol and kanamycin (50 µg/ml) for 48 h at 18 °C. Bacteria were pelleted and homogenized using EmulsiFlex and protein extracts were subjected to purification. Depending on proteins' stability, several various methods were used in purification procedures. 6xHis-SUMO-tagged proteins purification included following steps: Ni affinity chromatography on the 5 ml column filled with Ni-NTA Superflow resin (Qiagen), followed by SUMO protease on-column cleavage, desalting, a second round of Ni affinity chromatography with collection of unbound material, and gel filtration Hiload 16/60 Superdex S200 column (GE Healthcare). The 6xHis tagged proteins purification was simpler and included Ni affinity chromatography followed by gel filtration. The purification procedure was performed using an ÄKTA express apparatus. Purified proteins were analyzed by standard SDS-PAGE and Nanodrop (Thermo Fisher Scientific). When necessary additional ion exchange chromatography purification step was performed using 6 ml Resource Q column (GE Healthcare).

**Data availability**. RNA-seq data obtained in this study are available in the GEO repository: GSE106368. The mass spectrometry proteomics data have been deposited to the ProteomeXchange Consortium via the PRIDE[68] partner repository with the dataset identifier PXD009826. Other relevant data are included in the paper and accompanying Supplementary Information or are available from the corresponding author upon reasonable request.

All scripts used in bioinformatics analyses are available for researchers upon reasonable request.

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

## Acknowledgements

We thank the ENCODE Consortium, particularly members of Gene Yeo's group (UCSD) for sharing eCLIP data. We thank Dorota Adamska, Katarzyna Kowalska, and Kamila Klosowska-Kosicka for technical support, and Teresa Szczepinska for help with initial RNA-seq analyses. We thank Zbigniew Warkocki for discussions about biochemical experiments, Krzysztof Skowronek for help with circular dichroism experiments, and Sylwia Czarnomska for critical reading of the manuscript. We are grateful to Jeffrey Wilusz for sharing anti-GRSF1 antibodies. This work was mainly supported by a grant from the National Science Centre, Poland (UMO-2014/12/W/NZ1/00463 to R.J.S.) and co-supported by grants from the same institution (UMO-2013/11/13/NZ1/00089 to P.P.S., UMO-2014/13/D/NZ2/01114 to R.J.S.) and by ERC Starting Grant 309419 PAPs & PUPs (to A.D.). Experiments were carried out with the use of CePT infrastructure financed by the European Union: the European Regional Development Fund (Innovative economy 2007–13, Agreement POIG.02.02.00-14-024/08-00) and the Next Generation Sequencing Platform at the International Institute of Molecular and Cell Biology in Warsaw (no. 6405/IA/1789/2014).

## Author contributions

R.J.S. supervised the project. Z.P. performed biophysical assays, RNA-seq–G4RNA correlation analysis, and phylogenetic analysis, and also performed structural analysis and designed GRSF1 and tRNA-like mutants. M.A.W. and Z.P. purified proteins and performed biochemical experiments. L.S.B. performed experiments involving fluorescence microscopy. L.S.B., M.S., and P.P.S. verified RNA-seq data. T.M.K. carried out the bioinformatic analysis of RNA-seq and eCLIP data. R.J.S. and L.S.B. prepared RNA-seq libraries and performed co-purification studies. D.C. performed label-free quantification of M.S. data which were subsequently analyzed by R.J.S. R.J.S. drafted the manuscript. R.J.S. and Z.P. wrote the final version of the manuscript with contribution from A.D. and P.P.S. All authors participated in writing the Methods section and drawing final conclusions. R.J.S. and A.D. conceived the project.

## Additional information

**Competing interests:** The authors declare no competing interests.

