## [Peer Review File · Nature Communications]

Reviewers' Comments:

Reviewer #1:

Remarks to the Author:

The manuscript by Pietras et al. presents interesting RNA abundance data, much of which reinforces previous research by this group and others. In particular, the basic importance of PNPase-Suv3 in mtRNA degradation was demonstrated very well in a previous publication from this group (Borowski et al, 2012). The extension to highlight the role of G-quadruplex formation raises a number of concerns, as follows:

1. While the authors are probably correct that suv3-PNPase is the major RNA-degrading activity in mammalian mitochondria, they should note on line 57 that other activities have been identified including LACTB2 (Levy et al NAR 44, 1813). Similarly, GRSF1 is not the only RRM-domain protein in mitochondria. At least one other is SLIRP. The authors might extend their analysis to this protein. The authors touch on SLIRP-LRPPRC in the discussion, but do not consider the independent RNA binding ability of SLIRP.
2. The authors use TAP-Tag pulldown to examine protein-protein interactions, leading to identification of a short list of interactors for Suv3 and PNPase including GRSF1. It does not seem that the complexes were treated to prevent co-IP of proteins associated indirectly through RNA interactions. Further support for direct protein interactions is warranted.
3. The GRSF1 staining in Fig 1B is weak and does not appear to support a strong co-localization with the BiFC fluorescence.
4. The RNA seq analysis of the effects of down-regulation of PNPase-SUV3 is generally a strong part of the manuscript. However, when extended to analysis of GRSF1, the authors state "As for degradosome-impaired cells, silencing of GRSF1 resulted in accumulation of L-strand transcripts". This seems misleading since silencing PNPase yielded up to 350 D-loop L-strand RNAs but GRSF1 silencing generated only 30.
5. The manuscript devotes a great deal of attention to the identification of a short RNA they call RNA-like since it has a 3' CCA modification. The primary results on this RNA are a bit confusing. In Fig 3A, the authors report hundreds of observations of this RNA when PNPase is silenced or with dominant-negative hSuv3, but the expanded view in 3b shows very few RNAs. This RNA is claimed to accumulate when GRSF1 is down-regulated, but it seems there are very few instances of it when GRSF1 is silenced (Fig 3c,d), and the Northern signal in Fig 3e is weak for the si GRSF1 lane. There does not seem to be any other intrinsic interest to this RNA. This might increase if the authors could show it is a conserved feature in vertebrate mtDNA genomes despite the drift in primary sequence of the mtDNA. To say as in line 184 that this is "THE in vivo substrate" (my caps) for GRSF1 and the degradosomes seems to exaggerate its importance. It does appear to be "an in vivo substrate"
6. The authors suggest (lines 219-20) that tRNA-like has G4-forming potential based on 3 pairs of adjacent Gs and a G triplet. The initial description of guanine tetrads suggested a requirement for 4 consecutive G residues for forming such a structure, and many papers have explored the potential for formation of such structures by shorter or interrupted G-stretches, with equivocal results. It is quite possible that the degradosomes has difficulty getting through G-rich sequences for some other reason, but a classical guanine tetrad is not necessarily involved here. This is reflected in the discussion in which the authors cite an analysis (ref 61) that most "predicted" G4s are unfolded in vivo. Some of the ones featured in this manuscript may not form quadruplexes.
7. In Fig 5a the authors show GRSF1 increases the fluorescence of TERRA RNA labeled with fluorophore and quencher. This assay seems straightforward. However, in b and c, they show GRSF reduces fluorescence in the ThT assay. It seems this could reflect only a binding competition between the protein and the ThT, not necessarily a specific melting phenomenon. Thus in line 253, the authors should conclude their results indicate the disruption or shielding of G4 structures.
8. Results in Fig 6 reinforce the importance of Suv3 for PNPase activity on some substrates, especially Mito3, but the general influence of GRSF1 on the reactions is not particularly strong. This detracts from the significance of the work.
9. The authors show a strong correlation of the presence of GRSF1 with the occurrence of RNA

sequences containing G4 sequences. This is noteworthy, but it is hard to impute a functional mechanism to such an evolutionary coincidence. This concern is supported by some of the data in table S3, where results for GRSF1-positive *Callorhinchus* and GRSF1-negative *Branchiostoma* seem similar.

Minor

1. Line 198, the meaning of "regular expression revealed" is unclear (it seems to be a computer string search strategy, but is not used correctly in this context)
2. Line 174 "only be" should be "only by"
3. Fig. 4 and S4: thermal denaturation panels misspell "wavelength" repeatedly
4. Line 405 the word on should be deleted from "can influence on the activity"

Reviewer #2:

Remarks to the Author:

In this ms, the authors describe a dedicated mechanism of degradation of G4 containing RNAs which prevents the accumulation of G4-containing transcripts in human mitochondria. Overall, this is an interesting manuscript with a lot of information. The hypothesis that the "evolutionary response to the increasing number of G4s is acquisition of a protein (GRSF1) that can melt such structures" is very interesting. I have a few specific queries:

§ starting line 237 The conclusion that "GRSF1 melts G4 RNAs" is based on FRET and ThT fluorescence. While I agree that this is the most likely conclusion, a word of caution would be welcome as artefacts are possible: binding of the protein may quench ThT and change the fluorescence emission of fluorescein.

Furthermore "the mutated GRSF1 protein could not unwind the TERRA substrate". This would suggest that the mutant binds but does not unwind, which is not case.

Figure 4a : define "regular expression" for G4 formation in the legend.

Line 210 and Figure 4b: briefly explain how 'stable' and 'unstable' G4 were defined. Reference 13 data refers to DNA; not RNA: assuming the stable G4 RNA correspond to the stable G4 DNA is perhaps dangerous.

Line 204: "RNA G4s are more stable than their DNA counterparts^{38,39,40}." : Most of the time! There are a few exceptions... a word of caution should be welcome => "generally more stable"

Lines 218-219: define this accumulating transcript.

Line 223: "ThT is specific for RNA G4". Again, a word of caution would be welcome. ThT also binds to DNA G4 and some other structures, including imperfect duplexes.

Figure 4E: Circular dichroism is probably not the best technique to establish G4 RNA formation: in contrast with DNA, RNA duplexes tend to have a CD signature resembling the one of RNA quadruplexes. TDS or UV-melting profiles at 295nm are actually more convincing. The tRNA-like oligoribonucleotide is unlikely to form a G4 based on the data shown in figure S4a.

Figure 7d: explain how G4 normalized to genome size is calculated.

Discussion: "Extraordinary GC skew of vertebrate mitochondrial genomes". A short discussion of GC skewness in genomic DNA would be welcome – are other regions with comparable skewness over 10-16kb?

Table S2: when listing a reference "modified", define what modification is considered.

Typos / details:

in Table S3: use dots, not commas for decimals. -0,41 => -0.41. Not sure all readers will understand what the authors mean by "aver 10k genomes".

Missing words in the sentence lines 173-174?

Reviewer #3:

Remarks to the Author:

The manuscript describes experiments that identify and functionally characterise a new protein partner of the human mitochondrial RNA degradosome. The degradosome is composed of the exoribonuclease polynucleotide phosphorylase and a RNA helicase.

The newly identified factor, GRSF1 interacts with the degradosome in vivo and in vitro, and evidence is presented that it can impact on the capacity of the assembly to degrade stably folded RNA formed by guanine tetraplexes. This is an interesting and important finding, and the evolutionary record appears to be consistent with a model in which the recruitment of the GRSF1 protein is correlated with GC skew on one strand of the mitochondrial DNA. The experimental and bioinformatics work is good and there are a few points that might be helpful for the authors to consider.

1. A key aspect of the paper is that a guanine tetraplex is the cognate target for the GRSF1-degradosome complex, but the data to prove that the RNA is in a tetraplex configuration appears to be qualitative. There is a formal possibility that the complex recognises another conformation.

2. Along a similar line, the data on the G melting behaviour of GRSF1 is qualitative. To what extent are the signals observed due to small conformational changes versus secondary structural change? The GRSF1 by itself may not be sufficient for melting the tetraplex, since the energies are likely to be very great. Another more quantitative readout might be hypochromic effect - is there a change in 260 nm signal in the presence of GRSF1?

Minor comments

Page 4, line 82 "...potential degradosome activators..." might be more accurate to state here instead "...potential degradosome partners..." since the procedure itself does not identify activators.

Page 4, line 87 "... and showed >3 mean enrichment..." The enrichment description is a little unclear on pages 14-15, lines 436. Is the intensity level for the protein not detected in mtTAP from the input sample?

Page 4, line 99 "Obtained results supported conclusion..." might read better as "The obtained results support the conclusion ..." Perhaps the past tense verb use here and in the next sentence might seem a little awkward?

Page 11, Discussion, line 336 Typo for G-quadruplexes

Page 11, Discussion, line 337 "Extraordinary GC skew..." might read better as "The extraordinary GC skew..."

Page 12, line 355 "In case of ..." might read better as "In the case of..."

Page 12, line 357 "Analysis of LRPPRC binding sites reveled very degenerate consensus recognition sequence." would read better as "Analysis of LRPPRC binding sites revealed a highly degenerate

consensus recognition sequence."

Page 12, line 359 "Such results corroborates with our data..." might read better as "Such results are in accord with our findings, ..."

Page 12, line 373 "...of the tRNA-like RNA".

Capital letters are used in the figure legends but lower case letters in the figures.

Supplementary figure S4 typo for wavelength for all figures

This is only question out of curiosity, but could modification of RNA impact on the potential tetraplex formation - oxidation of G or methylation of the sugar?

Have the authors looked at non-hydrolysable ATP analogs in the degradation experiments?

Reviewer #1

The manuscript by Pietras et al. presents interesting RNA abundance data, much of which reinforces previous research by this group and others. In particular, the basic importance of PNPase-Suv3 in mtRNA degradation was demonstrated very well in a previous publication from this group (Borowski et al, 2012). The extension to highlight the role of G-quadruplex formation raises a number of concerns, as follows:

1. While the authors are probably correct that suv3-PNPase is the major RNA-degrading activity in mammalian mitochondria, they should note on line 57 that other activities have been identified including LACTB2 (Levy et al NAR 44, 1813). Similarly, GRSF1 is not the only RRM-domain protein in mitochondria. At least one other is SLIRP. The authors might extend their analysis to this protein. The authors touch on SLIRP-LRPPRC in the discussion, but do not consider the independent RNA binding ability of SLIRP.

As suggested by the Reviewer we added a comment on other RNA degrading activities in the mitochondria (the discussion section, lines 388-393).

Thanks to the Reviewer's comment we realized that we should include more information on RRM proteins in the introduction section of the manuscript. RRM proteins have been divided into several groups (<https://www.ncbi.nlm.nih.gov/books/NBK63528/>, PMID: 18515081). GRSF1 belongs to the group of quasi-RRM proteins which form a distinct group within the RRM family as they bind RNA in a mode different from other family members; a phenomenon revealed at the structural level for the hnRNP F protein (PMID: 20526337). Importantly, this particular manner of RNA binding is incompatible with the G4 structure (Supplementary Fig. 6c) which suggests a molecular mechanism by which quasi-RRM proteins melt/prevent formation of G4 structures (PMID: 20526337). Two solved structures of GRSF1 qRRM domains in a complex with poly Gs RNA have been deposited in the PDB database. They adopt a very similar (1.3-2.0 Å RMSD) fold to the hnRNP F RRM domains (Supplementary Fig. 6a and b). Moreover, they bind poly-G RNA in the same manner as hnRNP F which is incompatible with folded G4 (Supplementary Fig. 6c). It seems that only possible outcome of GRSF1 binding G4 RNA is the disruption of the G4 structure.

SLIRP contains RRM motifs but it is not a quasi-RRM protein. Thus, our conclusion that GRSF1 is the only quasi-RRM protein in mitochondria is correct. However, we realized that we should highlight the difference between SLIRP (RRM) and GRSF1 (quasi-RRM) to avoid any confusion. This information was added to the revised version of the manuscript.

We did not consider the independent RNA binding ability of SLIRP as reports from Larsson and Filipovska labs indicate that SLIRP does not display LRPPRC-independent function (PMID: 27353330, PMID: 29146908). This was concluded based on *in vitro* binding assays (PMID: 27353330) and high-throughput identification of the *in vivo* SLIRP footprint (PMID: 29146908). In fact, the RRM domain of SLIRP was revealed to be responsible for binding to LRPPRC (PMID: 27353330). Moreover, SLIRP was shown to have very weak RNA binding properties on its own (PMID: 27353330). CLIP analysis for SLIRP protein indicated enrichment of mitochondrial transcripts but further experiments attributed their binding to LRPPRC activity in the LRPPRC-SLIRP complex (PMID: 27353330). In addition, mtRNAs which were enriched in this CLIP were products of the transcription of the mitochondrial H-strand. These RNAs are very unlikely to form G4 structures as our analysis (Fig. 4a) and the results of others (PMID: 26792894) did not reveal sequences in these transcripts which would be capable of forming the G4 structure. Altogether, we think that it is very unlikely that SLIRP alone can contribute to the G4 RNA surveillance mechanism which we describe here. Therefore, we did not include SLIRP analysis in our studies. Instead, we discuss LRPPRC-SLIRP complex, which was shown to bind mitochondrial mRNAs. Thus, we can speculate that G-poor sequences, which are enriched in mt-mRNAs are protected from degradation by LRPPRC-SLIRP complex while G-rich non-coding RNA species are degraded with the help of GRSF1.

2. The authors use TAP-Tag pulldown to examine protein-protein interactions, leading to identification of a short list of interactors for Suv3 and PNPase including GRSF1. It does not seem that the complexes were treated to prevent co-IP of proteins associated indirectly through RNA interactions. Further support for direct protein interactions is warranted.

Indeed, protein lysates were not treated with RNase before TAP-tag pulldown. For this reason we did not conclude that Suv3 and PNPase interact directly with GRSF1. We stated in the manuscript that they co-purify or associate. Importantly, our main conclusions are correct regardless whether GRSF1-degradosome interaction is direct or mediated by RNA. The role of GRSF1 is to affect G4 structures which enables efficient RNA degradation by the degradosome. To fulfill this function GRSF1 does not need to interact directly with the degradosome.

Nevertheless, we tested whether the interaction is mediated by RNA. To this end we repeated TAP-tag pulldown and found that a lower amount of GRSF1 co-purifies with the degradosome components if protein lysates are treated with RNase. This indicates that interaction of GRSF1 with the degradosome is RNA dependent. This new data is presented as Supplementary Fig. 1a.

3. The GRSF1 staining in Fig 1B is weak and does not appear to support a strong co-localization with the BiFC fluorescence.

Although the GRSF1 signal was not strong it was distinguishable from the background which let us perform reliable co-localization analysis. Nevertheless, in order to avoid any confusion we repeated the experiment using other antibodies recognizing GRSF1. We obtained a stronger signal than previously which helped us to improve the corresponding figure (Fig. 1b, please note that in order to fulfill journal style recommendations we changed green and red colors into turquoise and magenta). Importantly, the new data led to the same conclusions as the old one. In addition, we confirmed that applied anti-GRSF1 antibodies specifically detect GRSF1 in immunofluorescence staining (new Supplementary Fig. 1b).

4. The RNA seq analysis of the effects of down-regulation of PNPase-SUV3 is generally a strong part of the manuscript. However, when extended to analysis of GRSF1, the authors state “As for degradosome-impaired cells, silencing of GRSF1 resulted in accumulation of L-strand transcripts”. This seems misleading since silencing PNPase yielded up to 350 D-loop L-strand RNAs but GRSF1 silencing generated only 30.

We agree that our initial description could be misleading. Our intention was to note that we observed accumulation of L-strand transcripts after GRSF1 silencing in comparison to control sample, similarly as in the case of down-regulation of PNPase or expression of dominant-negative version of Suv3. Indeed, the effect of GRSF1 silencing is not as strong as for the dysfunction of degradosome components. This part of the manuscript was changed to improve clarity.

5. The manuscript devotes a great deal of attention to the identification of a short RNA they call RNA-like since it has a 3' CCA modification. The primary results on this RNA are a bit confusing. In Fig 3A, the authors report hundreds of observations of this RNA when PNPase is silenced or with dominant-negative hSuv3, but the expanded view in 3b shows very few RNAs. This RNA is claimed to accumulate when GRSF1 is down-regulated, but it seems there are very few instances of it when GRSF1 is silenced (Fig 3c,d), and the Northern signal in Fig 3e is weak for the si GRSF1 lane. There does not seem to be any other intrinsic interest to this RNA. This might increase if the authors could show it is a conserved feature in vertebrate mtDNA genomes despite the drift in primary sequence of the mtDNA. To say as in line 184 that this is “THE in vivo substrate” (my caps) for GRSF1 and the degradosomes seems to exaggerate its importance. It does appear to be “an in vivo substrate”

Thanks to the Reviewer comment we realized that the expanded view shown in Fig. 3B was presented in a misleading way. In this figure we modified the scale to show that tRNA-like is also detectable in control samples but we neither described nor labelled that the level of tRNA-like in PNPase and Suv3 samples is far beyond the maximal value shown on the scale. We modified the figure legend to avoid confusion.

We agree that the increase of tRNA-like level after GRSF1 silencing is not as strong as in the case of cells with silenced PNPase or expressing dominant-negative version of Suv3. However, we think that the accumulation of tRNA-like when GRSF1 is down-regulated is evident. The number of normalized read

counts corresponding to tRNA-like in RNA-seq track of GRSF1 silenced sample is not high (Fig. 3c, d), however, it is clearly higher than in control samples (Fig. 3c, d). As the Reviewer suggested that the northern blot signal (Fig. 3e) is weak we repeated this analysis using a probe labelled to higher specific activity. We obtained results which led to the same conclusion as previous data. tRNA-like is hardly detectable in control samples (siNon-targeting) but is clearly detected in siGRSF1 sample.

We analyzed if tRNA-like is a conserved feature in vertebrate mtDNA genomes. We found that its sequence is not highly conserved. We modified the text in line 184 as suggested by the Reviewer.

6. The authors suggest (lines 219-20) that tRNA-like has G4-forming potential based on 3 pairs of adjacent Gs and a G triplet. The initial description of guanine tetrads suggested a requirement for 4 consecutive G residues for forming such a structure, and many papers have explored the potential for formation of such structures by shorter or interrupted G-stretches, with equivocal results. It is quite possible that the degradosome has difficulty getting through G-rich sequences for some other reason, but a classical guanine tetrad is not necessarily involved here. This is reflected in the discussion in which the authors cite an analysis (ref 61) that most “predicted” G4s are unfolded *in vivo*. Some of the ones featured in this manuscript may not form quadruplexes.

This comment of the Reviewer is unclear for us. Indeed, formation of G4 structures by non-canonical sequences was reported and our data indicate that tRNA-like is one such example. In our studies we defined G4 RNAs based on our sequence analysis as well as experimental studies performed by Bedrat et al (PMID: 26792894). Bedrat and colleagues performed sequence based prediction of G4 in mtDNA and subsequently experimentally tested whether the identified sequences form G4 structures. DNA oligonucleotides of corresponding sequences were subjected to several biophysical and biochemical tests (PMID: 26792894). As a result the Authors identified, with high confidence, G4 forming sequences, which we used in our analysis. Importantly, several studies showed that RNA, which has a sequence corresponding to G4 DNA is also likely to form G4 (PMID: 19572668, 19736017, 20670662, 20420470). Moreover, using different biophysical and biochemical tests we confirmed for selected RNAs, which are targeted by GRSF1 and the degradosome, that they form G4s. In addition, a sequence termed Mito3, for which we found that its degradation by the degradosome is facilitated by GRSF1 in the *in vitro* reconstituted system, was shown to form G4 by independent researchers (PMID: 20798345). Notably, GRSF1 belongs to quasi-RRM family for which the mechanism of binding and melting of G4 RNA structures was revealed at the structural level (PMID: 20526337). Analysis of structural data on GRSF1 indicates that this protein binds RNA in a manner which is incompatible with G4 formation (new Supplementary Fig. 6). Furthermore, the studies of McRae et al. who performed pull-down based search for proteins that bind G4 RNAs identified GRSF1 as one of top hits (PMID: 28472472). Altogether, this data indicates that GRSF1 targets G4 RNAs, thus, it is most likely the mechanism by which the protein enhances degradosome activity.

To support this conclusion further we examined whether GRSF1 augments degradosome-mediated degradation of G-rich substrates, which do not form G4 structure. To this end we introduced mutations into the Mito3 substrate which precluded formation of G4. We found that degradation of such RNAs by the degradosome is not augmented by GRSF1, although guanine is the most frequent nucleotide (please note that it is difficult to design very G-rich substrates which do not form G4) (new Supplementary Fig. 9). Importantly, we experimentally confirmed that mutated Mito3 substrates do not form G4 (new Supplementary Fig. 9a). Altogether, we think that our results strongly support the conclusion that RNAs which are subjects of cognate action of GRSF1 and the degradosome form G4 structures.

The Reviewer refers to reference 61 (PMID: 27708011). Importantly, this work does not undermine capability of “predicted” G4 RNAs to acquire this structure. The Authors of this paper propose that either an organism does not have G4 sequences (e.g., prokaryotes) or it developed countermeasures to preserve the majority of these G4 RNAs in an unfolded state (eukaryotes). A mechanism, which we reveal in the manuscript, is in agreement with hypothesis put forward by Guo and Bartel (mentioned Ref. 61) that eukaryotes have protein machinery which controls G4 RNAs. Co-operation of GRSF1 and the degradosome enables efficient elimination of mitochondrial G4 RNAs, which involves G4 melting (GRSF1) and RNA degradation (degradosome).

7. In Fig 5a the authors show GRSF1 increases the fluorescence of TERRA RNA labeled with fluorophore and quencher. This assay seems straightforward. However, in b and c, they show GRSF reduces fluorescence in the ThT assay. It seems this could reflect only a binding competition between the protein and the ThT, not necessarily a specific melting phenomenon. Thus in line 253, the authors should conclude their results indicate the disruption or shielding of G4 structures.

We applied a ThT assay as the dye was well-documented for detection of G4 structures and enables fast testing of many substrates. We agree that a ThT assay alone would not be sufficient to conclude that GRSF1 melts G4 structures. For this reason we applied a different approach involving TERRA RNA labeled with a fluorophore and quencher. Results of this assay confirmed our conclusion. Moreover, in the revised version of the manuscript we included results of another assay based on different principles. This new data strengthens the initial conclusion that GRSF1 melts G4 structures (Fig. 5e). Nevertheless, in agreement with the Reviewer's comment we modified the description of ThT data to indicate the fact that a ThT assay alone is not sufficient to draw conclusion about melting.

8. Results in Fig 6 reinforce the importance of Suv3 for PNPase activity on some substrates, especially Mito3, but the general influence of GRSF1 on the reactions is not particularly strong. This detracts from the significance of the work.

In our opinion results in Fig. 6 support the importance of GRSF1 as well. In the presence of GRSF1, degradation of substrates by the degradosome was 1.75 or 1.5-fold more efficient than in the absence of GRSF1. The degradation assays which we presented in the initial submission were analyzed at the time point when a significant fraction of substrates was already degraded (90% in the case of Mito3), thus, comparison of reaction rates might be affected. Therefore, we repeated the degradation assay in which we collected reaction products after a shorter incubation time (new Supplementary Fig. 8). This analysis showed stronger influence of GRSF1 on degradosome activity (3-fold).

Altogether, we think that results of *in vitro* RNA degradation assays are in agreement with our conclusion that GRSF1 has an auxiliary role in mtRNA G4 degradation. Notably, from the evolutionary point of view the degradosome appeared in mitochondria earlier than GRSF1. Our phylogenetic analysis indicates that GRSF1 was acquired to mitochondria when mitochondrial genomes evolved into molecules encoding many RNAs prone to form G4 structures. Therefore, we can speculate that it is not surprising that the degradosome has an ability to degrade G4 as it is supposed to deal with G4 RNAs also in GRSF1-negative organism. GRSF1 and its auxiliary role become important when mitochondrial genomes become G4 rich since the activity of the degradosome alone was not sufficient to degrade so many of them.

9. The authors show a strong correlation of the presence of GRSF1 with the occurrence of RNA sequences containing G4 sequences. This is noteworthy, but it is hard to impute a functional mechanism to such an evolutionary coincidence. This concern is supported by some of the data in table S3, where results for GRSF1-positive *Callorhinchus* and GRSF1-negative *Branchiostoma* seem similar.

This part of our studies was particularly appreciated by two other Reviewers. In general it is rare when evolutionary conclusions can be fully verified experimentally. The fact that we cannot modify/engineer the mitochondrial genome precludes further examinations. Nevertheless, we believe that results of our analyses strongly suggest that GRSF1 was acquired to mitochondria when the number of G4 forming sequences exceeded the capacity of protein machinery, which had been responsible for surveillance of these structures before acquisition of GRSF1. The Reviewer indicated two organisms, which seem to stand an exception from the rule. We aimed for our analysis to be thorough; therefore we included species as close to the vertebrate/invertebrate boundary as possible. *Callorhinchus* and *Branchiostoma* are such cases. The results are similar between those organisms as they are probably close to the shift from GRSF1-negative to GRSF1-positive and G4-poor to G4-rich species and could represent a transitional stage. However the differences between genomes of GRSF1-negative and GRSF1-positive organisms are clear and highly significant as demonstrated on Fig. 7d. This is not the first case when mitochondrial genomes exhibit some exceptions. For example, vertebrate mitochondrial genomes have almost the same organization in all taxa, however, avian mitochondrial genomes do not possess oriL which in other vertebrates functions as origin of

replication. Altogether, although we are not able to perform direct functional experiments, we think that the strong correlation, which we observe has functional implication and propose that GRSF1 was acquired to mitochondria to facilitate degradation of G4 RNAs.

Minor

1. Line 198, the meaning of “regular expression revealed” is unclear (it seems to be a computer string search strategy, but is not used correctly in this context).

This was corrected in the revised version of the manuscript.

2. Line 174 “only be” should be “only by”

This was corrected as suggested by the Reviewer.

3. Fig. 4 and S4: thermal denaturation panels misspell “wavelength” repeatedly

This was corrected as suggested by the Reviewer.

4. Line 405 the word on should be deleted from “can influence on the activity”

This was corrected as suggested by the Reviewer.

Reviewer #2

In this ms, the authors describe a dedicated mechanism of degradation of G4 containing RNAs which prevents the accumulation of G4-containing transcripts in human mitochondria. Overall, this is an interesting manuscript with a lot of information. The hypothesis that the "evolutionary response to the increasing number of G4s is acquisition of a protein (GRSF1) that can melt such structures" is very interesting. I have a few specific queries:

§ starting line 237 The conclusion that "GRSF1 melts G4 RNAs" is based on FRET and ThT fluorescence. While I agree that this is the most likely conclusion, a word of caution would be welcome as artefacts are possible: binding of the protein may quench ThT and change the fluorescence emission of fluorescein. Furthermore "the mutated GRSF1 protein could not unwind the TERRA substrate". This would suggest that the mutant binds but does not unwind, which is not case.

We applied a ThT assay as the dye is well-documented in detection of G4 structures and enables fast testing of many substrates. We agree that a ThT assay alone would not be sufficient to conclude that GRSF1 melts G4 structures. For this reason we applied a different approach involving TERRA RNA labeled with a fluorophore and a quencher. Importantly, we found different response of the substrate to GRSF1 depending on ions present in the mixture. Increase of fluorescence, an indicative of substrate melting, was smaller when G4-stabilizing K^+ ions were present in the reaction. To exclude the possibility suggested by the Reviewer that binding of GRSF1 may change the fluorescence emission of fluorescein we tested fluorescence of TERRA RNA labeled only with the fluorophore. We found that GRSF1 does not enhance emission of the fluorescence (new Supplementary Fig. 5c), thus, it is unlikely that the effect which we observed upon binding of GRSF1 to doubly labelled substrate is caused by changes in fluorescein properties. To further support our conclusion that GRSF1 melts G4 structures we performed another assay which is based on different principles than the ones mentioned above. This new data presented on Fig. 5e also indicated that GRSF1 disrupts G4 structures. Overall we think that based on results of these three different approaches we can conclude that GRSF1 melts G4 RNAs. In agreement with the Reviewer suggestion we added a word of caution about ThT assay.

In agreement with the Reviewer's suggestion we changed the sentence “the mutated GRSF1 protein could not unwind the TERRA substrate” to avoid any ambiguity.

Figure 4a : define "regular expression" for G4 formation in the legend.

This was defined as suggested by the Reviewer.

Line 210 and Figure 4b: briefly explain how "stable" and "unstable" G4 were defined. Reference 13 data refers to DNA; not RNA: assuming the stable G4 RNA correspond to the stable G4 DNA is perhaps dangerous.

Because studies of equivalent RNA and DNA sequences showed that RNA versions of G4 DNA could also form G4s we applied classification of mitochondrial G4 DNAs (reference 13) to their RNA counterparts. It seems that such extrapolation might be inaccurate. Thus, we removed from the manuscript results which took this classification into account and presented results of the analysis which concerned all G4 sequences without their sub-classification. Importantly, this analysis led to the same conclusion as the one with stable and unstable G4. We found accumulation of RNAs upstream of G4s when function of degradosome or GRSF1 was impaired which indicates that under these conditions transcripts degradation cannot proceed efficiently across the G4 sequences.

Line 204: "RNA G4s are more stable than their DNA counterparts^{38,39,40}." Most of the time! There are a few exceptions... a word of caution should be welcome => "generally more stable"

This was changed as suggested by the Reviewer.

Lines 218-219: define this accumulating transcript.

Lines 218-219 refer to tRNA-like which was defined in lines 162-167 of the initial manuscript. We modified lines 218-219 to improve clarity.

Line 223: "ThT is specific for RNA G4". Again, a word of caution would be welcome. ThT also binds to DNA G4 and some other structures, including imperfect duplexes.

We agree with the Reviewer that indicated phrase can be misleading. Therefore, we modified the text to avoid impression that ThT can bind only G4 RNAs. Importantly, in our experiments we included samples that contained non-G4 forms of RNA (dsRNA, ssRNA and GNRA - a hairpin consisting of eight GC base pairs and GCAA tetraloop (PMID: 1712983, PMID: 19111177)). In agreement with the results of others (PMID: 27098781) we found that the presence of G4 RNAs results in much higher fluorescence of ThT in comparison to other tested RNA species.

Figure 4E: Circular dichroism is probably not the best technique to establish G4 RNA formation: in contrast with DNA, RNA duplexes tend to have a CD signature resembling the one of RNA quadruplexes. TDS or UV-melting profiles at 295nm are actually more convincing. The tRNA-like oligoribonucleotide is unlikely to form a G4 based on the data shown in figure S4a.

Since tRNA-like is much longer than other examined substrates there is higher chance that it can adopt more than one structure. We added a comment in the result section to indicate that this RNA does not respond in all assays as canonical G4 RNA.

Figure 7d: explain how G4 normalized to genome size is calculated.

This information is included in the revised version of the manuscript. The normalization to genome size was done mainly to exclude artificial outlier, which originated from species of scallop that has unusually large mitochondrial genome (32 kb), which is twice the usual size.

Discussion: "Extraordinary GC skew of vertebrate mitochondrial genomes". A short discussion of GC skewness in genomic DNA would be welcome – are other regions with comparable skewness over 10-16kb? Unfortunately, due to space constraints we could not add suggested discussion.

To answer the Reviewer's question we analyzed GC skewness of human nuclear genome. To this end the sliding window approach was applied using 16 kb window and 0.5 kb window step. Among such generated regions only 0.08% have GC skewness equal or higher than human mitochondrial genome (figure included for the Reviewer). Moreover, we found that these regions show low conservation among 100 vertebrates (full list of organisms which nuclear genomes were included in the conservation analysis can be found here: http://genome.ucsc.edu/cgi-bin/hgTrackUi?hgsid=653656745_ikQWpbWIyg9m6agmow75ltJA8cIG&c=chr1&g=cons100way) which may suggest that they do not have functional importance.

Table S2: when listing a reference "modified", define what modification is considered.

Our intention was to indicate that a sequence of a given oligonucleotide was changed and differs from the one present in the reference. This was improved in the revised version of the manuscript.

Typos / details:

in Table S3: use dots, not commas for decimals. -0,41 => -0.41. Not sure all readers will understand what the authors mean by "aver 10k genomes".

Commas were replaced with dots. Description of "aver 10k genomes" was improved.

Missing words in the sentence lines 173-174?

Indeed, we missed "but not with other probes". This was corrected in the revised version of the manuscript.

Reviewer #3

The manuscript describes experiments that identify and functionally characterise a new protein partner of the human mitochondrial RNA degradosome. The degradosome is composed of the exoribonuclease polynucleotide phosphorylase and a RNA helicase. The newly identified factor, GRSF1 interacts with the degradosome *in vivo* and *in vitro*, and evidence is presented that it can impact on the capacity of the assembly to degrade stably folded RNA formed by guanine tetraplexes. This is an interesting and important finding, and the evolutionary record appears to be consistent with a model in which the recruitment of the GRSF1 protein is correlated with GC skew on one strand of the mitochondrial DNA. The experimental and bioinformatics work is good and there are a few points that might be helpful for the authors to consider.

1. A key aspect of the paper is that a guanine tetraplex is the cognate target for the GRSF1-degradosome complex, but the data to prove that the RNA is in a tetraplex configuration appears to be qualitative. There is a formal possibility that the complex recognises another conformation.

Several lines of evidence support the conclusion that G4 is the cognate target for GRSF1 and the degradosome. **First**, a group of mtRNAs accumulating upon GRSF1 or degradosome dysfunction includes transcripts which can form G4 but not their complementary counterparts, which are unlikely to form G4. In addition, our bioinformatic analysis revealed that reduction of GRSF1 levels or degradosome activity leads to accumulation of RNAs upstream of G4s, which indicates that under these conditions transcripts degradation cannot proceed efficiently across the G4 sequences. This result is in agreement with biochemical data on the degradosome components, which act from the 3' to the 5' end of RNA.

Second, GRSF1 belongs to quasi-RRM family for which mechanism of binding and melting of G4 RNA structures was revealed (PMID: 20526337). In agreement with this, our *in vitro* binding assays as well as results of CLIP-experiments performed by others (PMID: 22955616, PMID: 25683715) showed that GRSF1 binds RNAs that adopt G4 structures. Furthermore, recent studies of McRae et al, who performed pull-down search for proteins binding G4 RNA, identified GRSF1 as one of top hits (PMID: 28472472). Thus, ability of GRSF1 to bind G4 RNAs is supported by various data.

Third, data included in the initial version of the manuscripts as well as new added results (see response to the next comment) indicate that binding of GRSF1 to G4 containing RNA affects G4 which is associated with enhanced degradation of this RNAs by the degradosome. The latter conclusion was strengthened in the revised version of the manuscript by an experiment in which we used a version of Mito3 substrate which was modified as not to form G4 structure. Results of our three experiments, which employ different principles, as well as results obtained by others (PMID: 20798345, PMID: 26792894) showed that Mito3 sequence forms G4. We designed two versions of Mito3 substrate in which several Gs were substituted to abolish formation of G4. We found that GRSF1 has no effect on degradation of mutated versions of Mito3 by the degradosome (new Supplementary Fig. 9). These results further support the conclusion that G4 structure is the cognate target of GRSF1 and the degradosome. They are also in line with a conclusion that the function of GRSF1 is to maintain RNA in non-G4 form.

Altogether, we think that our data strongly support the conclusion that G4 structures are the targets of cooperation between GRSF1 and the degradosome. Although, formally it is possible that in addition to G4 structures the cooperation can be also important for degradation of other RNA structures we prefer not to

speculate on them in the manuscript as their identity is unclear and we think that our original conclusion is supported by multiple lines of evidence.

2. Along a similar line, the data on the G melting behaviour of GRSF1 is qualitative. To what extent are the signals observed due to small conformational changes versus secondary structural change? The GRSF1 by itself may not be sufficient for melting the tetraplex, since the energies are likely to be very great. Another more quantitative readout might be hypochromic effect - is there a change in 260 nm signal in the presence of GRSF1?

We thank Reviewer for excellent suggestion. We attempted to examine ability of GRSF1 to melt/disrupt G4 using hypochromic effect. Unfortunately, we faced some technical problems. While for RNA and DNA duplexes the difference in absorption due to hypochromic effect is around 25%, it is much lower for G4s (around 4%). This, and the fact that it was necessary to include protein in the measured sample, which also contributes to 260 nm readout, made that particular assay very challenging. As a result our attempts to record such effect were unsuccessful. However, we have followed similar line of thought and performed CD measurements. We have focused on 265 nm where maximal signal is recorded for folded RNA sample (well-defined G4 – Mito3) and the contribution from the protein is minimal. We have recorded decrease in 265 nm signal in presence of GRSF1, which is fully compatible with the idea that GRSF1 disrupts G4 structure (new data presented on Fig. 5e).

Additionally we performed analysis of available structural data on GRSF1, which indicates that the protein binds RNA in a manner, which is incompatible with G4 formation (new Supplementary Fig. 6). This suggests secondary structural change upon binding of GRSF1.

Minor comments

Page 4, line 82 "...potential degradosome activators..." might be more accurate to state here instead "...potential degradosome partners.." since the procedure itself does not identify activators.

This was changed according to Reviewer's suggestion.

Page 4, line 87 ".. and showed >3 mean enrichment..." The enrichment description is a little unclear on pages 14-15, lines 436. Is the intensity level for the protein not detected in mtTAP from the input sample?

Thanks to Reviewer comment we realized that the description on page 4 was confusing. We removed a part of the sentence which might have been confusing. In our proteomic experiment as potential biologically relevant interactors we considered proteins identified in all PNPase and Suv3 purifications, which level was clearly higher than in control mtTAP purifications. All of them were also present in control mtTAP purification, although at significantly lower levels. We have not performed mass spectrometry measurements on input samples

Page 4, line 99 "Obtained results supported conclusion..." might read better as "The obtained results support the conclusion ...". Perhaps the past tense verb use here and in the next sentence might seem a little awkward?

This was changed according to Reviewer's suggestion.

Page 11, Discussion, line 336 Typo for G-quadruplexes

This was changed according to Reviewer's suggestion.

Page 11, Discussion, line 337 "Extraordinary GC skew..." might read better as "The extraordinary GC skew..."

This was changed according to Reviewer's suggestion.

Page 12, line 355 "In case of .." might read better as "In the case of..."

This was changed according to Reviewer's suggestion.

Page 12, line 357 "Analysis of LRPPRC binding sites revealed very degenerate consensus recognition

sequence." would read better as "Analysis of LRPPRC binding sites revealed a highly degenerate consensus recognition sequence."

This was changed according to Reviewer's suggestion.

Page 12, line 359 "Such results corroborates with our data..." might read better as "Such results are in accord with our findings, ..."

This was changed according to Reviewer's suggestion.

Page 12, line 373 "..of the tRNA-like RNA".

This was changed according to Reviewer's suggestion.

Capital letters are used in the figure legends but lower case letters in the figures.

This was changed according to Reviewer's suggestion.

Supplementary figure S4 typo for wavelength for all figures

This was changed according to Reviewer's suggestion.

This is only question out of curiosity, but could modification of RNA impact on the potential tetraplex formation - oxidation of G or methylation of the sugar?

As G-quadruplex is formed with contribution from N1, N2, O6 and N7 of guanine base any modification of these would most likely result in disruption of tetrad formation. Currently most studies concerning modifications are focused on gene regulation and DNA modifications (PMID: 18447358, PMID: 28829124).

Have the authors looked at non-hydrolysable ATP analogs in the degradation experiments? We thank Reviewer for excellent suggestion. We have performed degradation assays with two ATP analogs and the results were similar to ones obtained in absence of the helicase hSuv3. This shows that hSuv3 not only have to be present but also have to be able to express its activity in order for G4s to be efficiently degraded. The results are presented in new Supplementary Fig. 10.

Reviewers' Comments:

Reviewer #1:

Remarks to the Author:

The revised version of manuscript NCOMMS-17-31751A is significantly improved as indicated in the detailed rebuttal and the revised text. I have only a few comments, none of which are particularly critical.

1. The description of degradosomes associated proteins fails to mention that C1QBP has been studied in this context before (Yagi, M., et al. (2012). "p32/gC1qR is indispensable for fetal development and mitochondrial translation: importance of its RNA-binding ability." *Nucleic Acids Research* 40(19): 9717-9737.)

2. I found the use of expanded insets in figure 3b to be confusing, where coverage of about 200-fold seems to decrease to an apparent 5-fold. The reader needs to note the explanation buried in the figure legend to the effect that the signal goes off scale. There must be a better way to represent this.

3. I continue to hold the opinion that the analysis of "tRNA-like" seems to be largely peripheral to the main focus of the paper. While this RNA, which has no apparent functional role, accumulates significantly when PNPase or Suv3 is silenced, it is barely detectable when GRSF1 is silenced in Figure 3. In addition the EMSA assays in Figure 5 show that it takes a very large concentration of GRSF1 (10-fold in excess of the RNA concentration) for appreciable binding to occur. I acknowledge that it is more readily detectable when GRSF1 is silenced, particularly in the new figure. Thus, while the consideration of this RNA is not a serious concern, it still seems to be a distraction.

4. An important conclusion of the paper is that GRSF1 enhances the rate of degradation of RNA by PNPase-Suv3. This is shown in Figure 6, but the results are not very dramatic. In the absence of GRSF1, about 40% of a substrate is degraded in 15 min while the addition of GRSF1 increases this to about 60%. There is also no indication that processive 3'-5' degradation stalls at a G4 structure in the absence of GRSF1. Have the authors challenged their enzymes with a longer RNA substrate containing a sequence with a strong G4 conformation potential, such as Mito3, in an isolated central region that would permit observation of a clear pause by PNPase in the absence of GRSF1?

Reviewer #2:

Remarks to the Author:

I am satisfied with the answers and modifications made by the authors. I recommend acceptance.

Detail (abstract, line 26):

"which lead to emergence of G4-containing RNAs"
seems too strong / implying a causality.

Would suggest changing it to:

"which allows the emergence of G4-containing RNAs"

Reviewer #3:

Remarks to the Author:

The authors have provided clear responses to the comments from all three reviewers and have modified the text and included additional data. The manuscript is much improved and the evidence for the model of the surveillance mechanism is stronger.

Response to Reviewers' comments.

We appreciate Reviewers's comments and are happy to hear that they found the revised version of the manuscript to be improved. It is nice to hear that 2 out of 3 Reviewers found the manuscript suitable for acceptance and that Reviewer #1 did not raise any critical issues.

Reviewer #1

The revised version of manuscript NCOMMS-17-31751A is significantly improved as indicated in the detailed rebuttal and the revised text. I have only a few comments, none of which are particularly critical.

1. The description of degradosomes associated proteins fails to mention that C1QBP has been studied in this context before (Yagi, M., et al. (2012). "p32/gC1qR is indispensable for fetal development and mitochondrial translation: importance of its RNA-binding ability." *Nucleic Acids Research* 40(19): 9717-9737.)

We added indicated reference as suggested by the Reviewer. Please note that in order to add this reference we had to remove another reference from the manuscript since the number of references is limited by the journal style.

2. I found the use of expanded insets in figure 3b to be confusing, where coverage of about 200-fold seems to decrease to an apparent 5-fold. The reader needs to note the explanation buried in the figure legend to the effect that the signal goes off scale. There must be a better way to represent this.

The figure was modified to improve clarity.

3. I continue to hold the opinion that the analysis of "tRNA-like" seems to be largely peripheral to the main focus of the paper. While this RNA, which has no apparent functional role, accumulates significantly when PNPase or Suv3 is silenced, it is barely detectable when GRSF1 is silenced in Figure 3. In addition the EMSA assays in Figure 5 show that it takes a very large concentration of GRSF1 (10-fold in excess of the RNA concentration) for appreciable binding to occur. I acknowledge that it is more readily detectable when GRSF1 is silenced, particularly in the new figure. Thus, while the consideration of this RNA is not a serious concern, it still seems to be a distraction.

Indeed, as commented by the Reviewer the biological role of tRNA-like is yet to be revealed. However, since this RNA species was overlooked in previous reports concerning mitochondrial RNA, and is an excellent marker of degradosome-mediated mtRNA decay capacity, we think that its description will be of great importance for the scientific community working on RNA metabolism in human mitochondria.

The Reviewer raised an issue concerning changes in tRNA-like levels, which seems to be the major reason why this part of the manuscript seems to be confusing. We believe that the Reviewer's confusion may be caused by the fact that the extent to which tRNA-like is upregulated upon GRSF1 silencing in RNA-seq experiments (Figure 3c-d) is different than the one observed in northern blot (Figure 3e, Figure 3f). Namely, the effect on the northern is stronger than that in the case of RNA-seq. It is a common situation that RNA-seq experiments show different extent of changes than hybridization-based assays. It was our mistake not to highlight this issue in the manuscript. We have added a comment in the revised version of the manuscript.

Regarding the EMSA. The typical EMSA experiments are conducted in such a way that increasing amounts of protein are mixed with a constant concentration of a substrate (RNA in our case). The presence of the shift is not depended on the RNA-protein ratio but the affinity. Since the shift is visible in the middle of the titration (Figure 5f) we are strongly convinced that the experiment is performed properly. Importantly, our binding reactions were performed in the presence of heparin as a competitor in order to ensure specificity of the observed reactions. The apparent K_d is in sub-micromolar range (around 100 nM), which ranks among

middle-strength binding, thus, the affinity of GRSF1 for tRNA-like is not weak which seems to be implied by the Reviewer comment.

Altogether, we think that identification and description of tRNA-like is an important part of our manuscript, supported by relevant data. Thus, we prefer to keep this data in the manuscript. However, we improved description of these results, particularly we emphasized the differences between RNA-seq and northern blot results.

4. An important conclusion of the paper is that GRSF1 enhances the rate of degradation of RNA by PNPase-Suv3. This is shown in Figure 6, but the results are not very dramatic. In the absence of GRSF1, about 40% of a substrate is degraded in 15 min while the addition of GRSF1 increases this to about 60%. There is also no indication that processive 3'-5' degradation stalls at a G4 structure in the absence of GRSF1. Have the authors challenged their enzymes with a longer RNA substrate containing a sequence with a strong G4 conformation potential, such as Mito3, in an isolated central region that would permit observation of a clear pause by PNPase in the absence of GRSF1?

We agree with the Reviewer that GRSF1 is not absolutely essential for decay of G4 forming RNAs. For this reason we concluded in the manuscript that GRSF1 facilitates or stimulates the ribonucleolytic capacity of the hSuv3-PNPase complex towards G4 containing substrates. We now modified the abstract to also reflect this conclusion. We did not challenge our enzymes with a longer RNA substrate.

Reviewer #2

I am satisfied with the answers and modifications made by the authors. I recommend acceptance.

Thank you!

Detail (abstract, line 26):

"which lead to emergence of G4-containing RNAs"
seems too strong / implying a causality.

Would suggest changing it to:

"which allows the emergence of G4-containing RNAs"

Modified as suggested by the Reviewer.

Reviewer #3

The authors have provided clear responses to the comments from all three reviewers and have modified the text and included additional data. The manuscript is much improved and the evidence for the model of the surveillance mechanism is stronger.

Thank you!